# Src-transformed cells hijack mitosis to extrude from the epithelium

Katarzyna A. Anton [1], Mihoko Kajita[2], Rika Narumi[2], Yasuyuki Fujita[2] & Masazumi Tada[1]

At the initial stage of carcinogenesis single mutated cells appear within an epithelium. Mammalian in vitro experiments show that potentially cancerous cells undergo live apical extrusion from normal monolayers. However, the mechanism underlying this process in vivo remains poorly understood. Mosaic expression of the oncogene vSrc in a simple epithelium of the early zebrafish embryo results in extrusion of transformed cells. Here we find that during extrusion components of the cytokinetic ring are recruited to adherens junctions of transformed cells, forming a misoriented pseudo-cytokinetic ring. As the ring constricts, it separates the basal from the apical part of the cell releasing both from the epithelium. This process requires cell cycle progression and occurs immediately after vSrc-transformed cell enters mitosis. To achieve extrusion, vSrc coordinates cell cycle progression, junctional integrity, cell survival and apicobasal polarity. Without vSrc, modulating these cellular processes reconstitutes vSrc-like extrusion, confirming their sufficiency for this process.

[1] Department of Cell & Developmental Biology, University College London, Gower Street, London WC1E 6BT, UK. [2] Division of Molecular Oncology, Institute for Genetic Medicine, Hokkaido University Graduate School of Chemical Sciences and Engineering, Sapporo 060-0815, Japan. Correspondence and requests for materials should be addressed to M.T. (email: m.tada@ucl.ac.uk)

At early stages of epithelial carcinogenesis, single mutations arise in single cells residing among normal neighbours within functioning organisms. In the past 10 years, several laboratories started uncovering a process called epithelial defence against cancer (EDAC)[1]. This is defined as a non-immunological primary defence mechanism whereby cells within an epithelial monolayer have the ability to sense and eliminate a mutated neighbour. Although recently the focus lied on the role of non-transformed neighbours in EDAC[2–4], there is evidence that transformed cells themselves have to undergo specific changes in the process of extrusion[5–8]. For example, in the case of vSrc-transformed cells (here referred to as vSrc cells), myosin activity regulated by myosin light chain kinase (MLCK) and Rho kinase (ROCK) as well as focal adhesion kinase (FAK) drive basal relocation of adherens junctions followed by apical extrusion[6]. Apart from mechanical shape adaptations, transformed cells residing among normal neighbours undergo changes in basic cellular functions that alter their metabolism[7] and endocytosis[8]. Until now, however, most studies of oncogenic cell extrusion have been performed using tissue culture models, cell lines and organoids, where cells are studied in environments different from the situation in vivo, such as matrix composed of just one protein, e.g. collagen I[6], or glass[9], a material of high rigidity. These culture conditions are known to affect cellular behaviour through modifying adhesion and cytoskeletal dynamics[10]. Since oncogenic cell extrusion requires complex rearrangements within a fully differentiated epithelium[11], it is important to investigate this phenomenon within a living organism, where cells can extrude and delaminate freely.

Here, we performed a comprehensive mechanistic study of oncogenic cell extrusion in vertebrate embryos of the zebrafish. Our model epithelium was the enveloping layer (EVL), the first polarised simple squamous epithelium that surrounds the yolk in the process of epiboly during gastrulation[12]. Unlike the *Drosophila* wing disc, the EVL is not prepatterned in the anteroposterior and dorsoventral axes[13], providing us with a homogenous cell population to study extrusion. Using the EVL-specific promoter Keratin18 (Krt18), we established a system in which the tamoxifen-inducible transcriptional activator Gal4 (KalTA4-ERT2) was expressed exclusively within the EVL[2,8] (Fig. 1a). In order to obtain mosaic expression of a given oncogene, we transiently injected constructs encoding oncogenes under the control of a UAS or double UAS element (dUAS driving bi-directional expression). We also created a double Krt18 promoter (dKrt18; Supplementary Fig. 1A, B) resulting in constitutive expression of modulators of extrusion within the EVL. Thus, this in vivo system allowed us to generate two discrete cell populations: transformed and normal cells in a differentiated homogenous tissue. This approach uncovered a novel mode of extrusion in which vSrc holds the cell in the G2 phase of the cell cycle until a misoriented pseudo-cytokinetic ring is formed and constricted in early mitosis, resulting in the cell leaving the epithelium.

## Results

**vSrc cells are apicobasally extruded from simple epithelia.** We previously showed that when the oncogene vSrc was mosaically expressed in the EVL, transformed cells were apically extruded (outside of the embryo)[6] (Fig. 1b, Supplementary Movie 1). Further careful investigation of this process through live-imaging revealed that transformed cells rounded up (became taller than their flat epithelial neighbours) and split into two fragments undergoing both apical and basal rather than exclusively apical extrusion (Fig. 1c, Supplementary Movie 2). Apical parts were always larger than basal parts of extruding cells. The size of basal parts released towards deep cells of the embryo varied from at least a third of the total cell

volume before extrusion to smaller basal vesicles (Fig. 1d). Extrusion rates were similar regardless of cluster sizes of vSrc cells, suggesting that this process is predominantly autonomous (Supplementary Fig. 2A). Since apoptosis could result in cell fragmentation[14], we investigated whether transformed cells died before becoming extruded. vSrc cells were negative for cleaved-Caspase-3 prior and following extrusion (Supplementary Fig. 2B). In contrast, EVL cells expressing death-associated kinase 1 (DAPK1) died before becoming basally extruded (Supplementary Fig. 2C). Moreover, inhibiting apoptosis by expressing the antiapoptotic protein XIAP[15] alongside vSrc, did not affect cell extrusion (Supplementary Fig. 2D). Following extrusion, larger apical parts of vSrc-transformed cells died, presumably via anoikis (Fig. 1c, Supplementary Movie 2). Therefore, we concluded that vSrc-mediated extrusion was not due to cell death.

**vSrc aberrantly regulates cytokinetic machinery in extrusion.** To investigate how vSrc cells produced apical and basal parts during extrusion, we analysed their properties. Live-imaging revealed that proliferation rates of vSrc cells within the EVL were significantly lower than control EVL cells expressing GFP only (Fig. 2a), likely because transformed cells were undergoing extrusion rather than mitosis (Fig. 2b). We then speculated that extruding vSrc cell used a contractile Actomyosin ring, reminiscent of the cytokinetic ring, for this separation. Visualising Actin and Myosin in vSrc-mediated extrusion was inconclusive, as both proteins are constitutively present at the cell cortex in ring-like structures coupled to adherens junctions (AJs)[16] (see Fig. 1b). To visualise a contractile Actomyosin ring, we focussed on Anillin, a scaffolding protein required for the assembly of the cytokinetic ring[17]. During cytokinesis, Anillin is recruited to the mitotic plane through active RhoA, which is localised there by signals from the mitotic spindle[18–20]. Anillin in turn recruits Myosin and Actin, orchestrating the assembly of a contractile ring[17]. Following constriction, Anillin localises to the midbody, and to the nucleus in the interphase[21]. In normal EVL cells, Anillin-GFP behaved as described[17,21,22] (Supplementary Fig. 3A, Supplementary Movie 3). However, in vSrc-expressing cells, Anillin was recruited to junctional foci in the lateral region and eventually formed a junctional ring, in addition to its nuclear localisation (Fig. 2c, Supplementary Movie 4). This ring constricted during extrusion and in the moment of apicobasal separation resembled the midbody, apparently inherited by the apical part of the extruding cell (Fig. 2c). Importantly, the junctional localisation of Anillin has been reported in cultured cells and in *Xenopus* embryos, where it is involved in junctional maintenance[23,24]. However, we did not observe junctional Anillin in normal EVL cells, with the exception of extrusion in vSrc cells and briefly in mitosis following nuclear envelope breakdown (NEB) before recruitment to the mitotic plane.

The Anillin ring in vSrc cells appeared to be contractile during extrusion, as Anillin-GFP colocalised with phospho-myosin light chain (pMLC) (Supplementary Fig. 3B). To confirm that Anillin was necessary for extrusion, we used Anillin morpholino[22]. Since Anillin abrogation delayed epiboly progression, we compared morpholino-treated embryos with or without autonomously supplied morpholino-resistant Anillin. With exogenous Anillin, extrusion rates were significantly higher (Fig. 2d). Additionally, we have created a dominant negative form of Anillin as described in *C. elegans*[25] containing the anillin and pleckstrin homology domains with the RhoA-binding motif, but lacking the myosin-binding and actin-binding domains. When co-expressed with vSrc, this form of Anillin significantly increased the rate of failed extrusions, during which cells became taller, rounder, but then reintegrated into the monolayer (Fig. 2e). Notably, unlike most

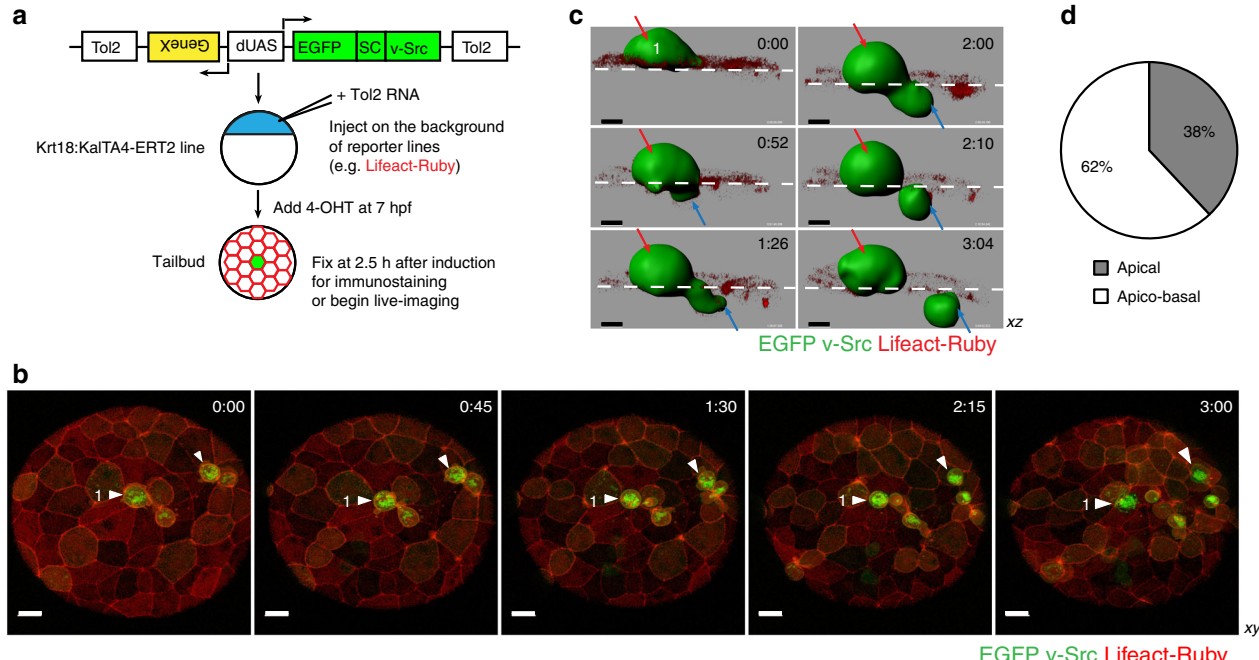

**Fig. 1** vSrc-transformed cells become apicobasally extruded from the EVL layer. **a** Experimental design. Fish embryos obtained from a transgenic line expressing tamoxifen-inducible Gal4 specifically in the EVL (Krt18:KalTA4-ERT2) are injected at one-cell stage with constructs encoding oncogenes and effectors/markers under the control of the bi-directional UAS, dUAS[8]. At 50–70% epiboly, embryos are treated with tamoxifen to induce oncogene expression. At tailbud (2–3 h from induction, 10 h post fertilization), embryos are fixed for quantification or mounted in agarose for live-imaging. **b** Time-lapse imaging of vSrc cell extrusion from the EVL of the zebrafish embryo. Transgenic embryos obtained from a line expressing an RFP-actin marker (red) specifically in the EVL (Krt18:Lifeact-Ruby) line crossed with the Krt18:KalTA4-ERT2 line were injected with the UAS:EGFP-vSrc construct (green). Movies were taken over 4 h. Frames were extracted from a representative movie at indicated times from the tailbud stage ($t = 0$). White arrowheads indicate cells that become extruded. "1" indicates the cell segmented in **c**. Scale bar, 30 μm. **c** Segmented time-lapse images of vSrc cell extrusion. The surface function was used to segment the GFP-positive vSrc cell from **b** over time using the Imaris software. In this cross section of the embryo (xz view), the cell is undergoing an apicobasal split (apical part extruding outside of the embryo is marked with red arrow and the basal part extruding towards the deep cells with blue arrow). The dashed white line indicates the surface of the embryo. Scale bar, 10 μm. **d** Quantification of vSrc-mediated cell extrusion type based on time-lapse imaging. Apicobasal extrusion refers to extrusions in which the basal part contains at least a third of the original cell volume. Seven embryos were imaged in seven independent experiments (total number of extrusions: $n_{Src} = 19$)

cultured cells[26], EVL cells do not normally undergo mitotic rounding (e.g. Supplementary Fig 3A), therefore rounding-up indicates early stages of extrusion. Interestingly, the Anillin ring itself was initially parallel to the plane of the epithelium. As the vSrc cell became taller, the ring shifted its orientation towards an ever more oblique position to the surface of the embryo before separation (Fig. 2f). Together, these data suggest that vSrc cells aberrantly utilise the cytokinetic machinery during extrusion.

**vSrc-driven extrusion occurs in early mitosis**. Since vSrc can modulate the cytokinetic machinery, the question remained whether vSrc-driven extrusion requires mitotic entry. To investigate the role of cell cycle progression (Fig. 3a) in extrusion, we used fixed embryos, therefore all the following extrusion rates represent combined scores of tall and apically extruded cells as these remain on the surface of the embryo, while basally extruded parts move away. Expression of the G2/M transition inhibitors Wee1 (Fig. 3b) or constitutively active Protein phosphatase 2 A (Pp2A) (Supplementary Fig. 4A) alongside vSrc led to strong inhibition of extrusion. Conversely, expression of a constitutively active form of the G2/M promoter Cdc25 (CA-Cdc25) increased extrusion (Fig. 3c). Wee1-mediated suppression of vSrc cell extrusion was rescued by constitutively active Cdk1 (Fig. 3d). These results were confirmed by G2/M arrest achieved with abrogation of the early mitotic inhibitor Emi1[27] (Supplementary Fig. 4B, C). These data suggest that transformed cells have to enter mitosis to extrude.

To establish whether extrusion is a misoriented mitosis, we imaged multiple mitotic markers in extruding vSrc cells. Visualisation of microtubules with Doublecortin (Dcx-GFP; Supplementary Fig. 4D) and chromatin with Histone 2B (H2B-GFP; Fig. 3e) revealed that prior to extrusion the mitotic spindle was not assembled and full NEB did not occur. As the nuclear volume grows throughout the cell cycle and peaks before mitosis[28], we measured the chromatin volume in cells approaching mitosis or extrusion by determining the volume of the positive H2B-GFP signal. We found that there was a set chromatin volume between 600 and 700 μm³ at which EVL cells expressing myr-Cherry (membrane marker) or vSrc divide, and that the same volume was reached by vSrc cells immediately before extrusion (Fig. 3f). Remarkably, the nucleus was always inherited by the larger apical part of the cell. We did not observe full NEB in extruding cells, although increased permeability of the nuclear envelope was detected; immediately before and after extrusion a portion of H2B-GFP and nuclear GFP was present in the cytoplasm (Fig. 3e, Supplementary Movies 5 and 6 and Supplementary Fig. 4E). Overall, these observations indicate that most vSrc-transformed cells in the EVL become extruded in a cell cycle-dependent manner instead of completing mitosis.

**Src activation results in G2/M arrest leading to extrusion**. To understand how Src activation modulates cell cycle progression, we established a transgenic line expressing Cyclin B1-GFP (CcnB1-GFP) in the EVL. Cyclin B1 forms a complex with

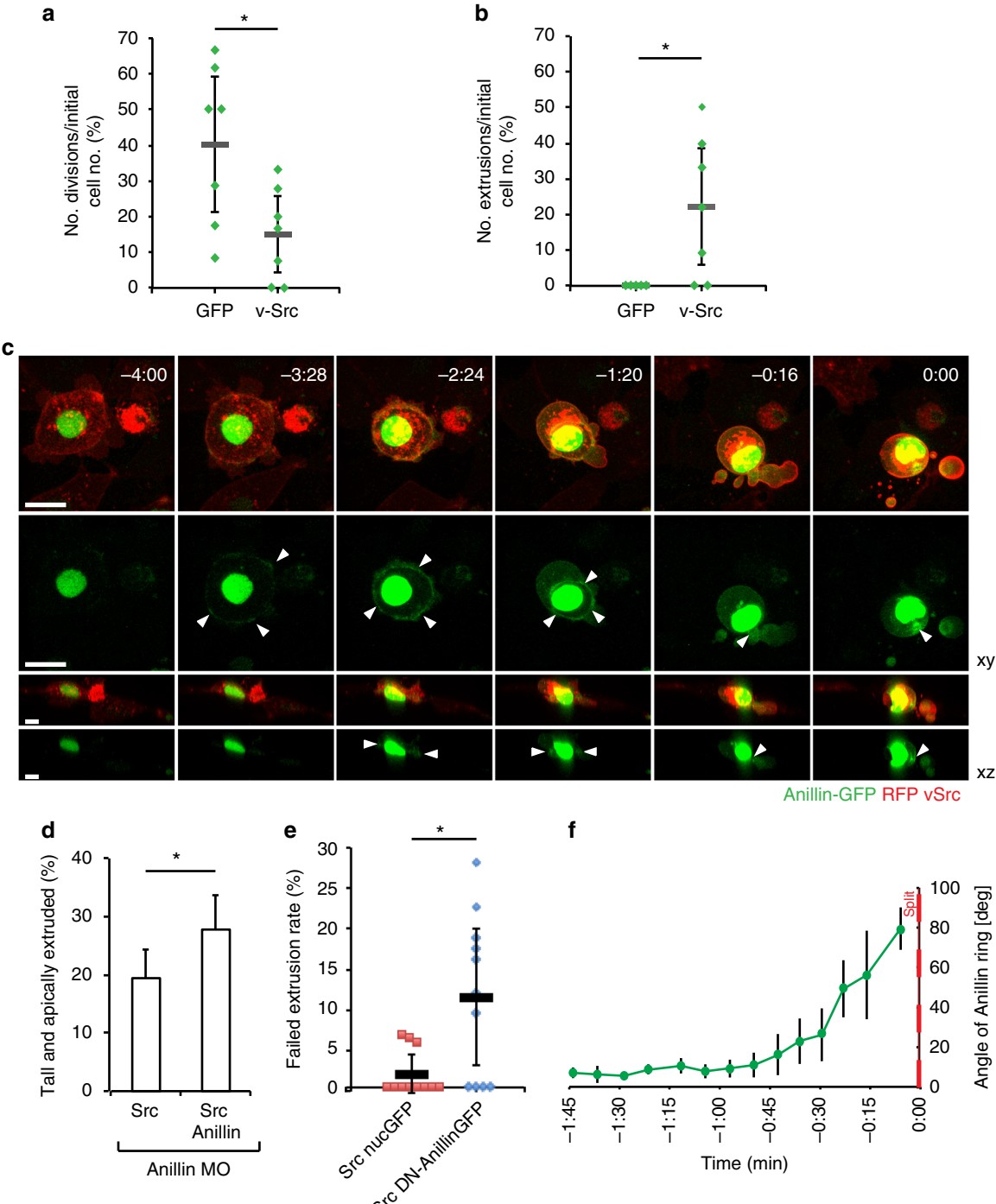

**Fig. 2** Contractile Anillin ring is recruited to the lateral cortex prior to extrusion. **a**, **b** Quantification of EGFP and EGFP-vSrc cell division (**a**) and extrusion (**b**) rates based on time-lapse imaging. Division rates were calculated as the number of divisions over 4 h divided by the initial number of GFP-positive cells per embryo. Each mark represents a division rate in a single embryo. Grey lines represent mean values ± standard deviation (s.d.). Seven embryos were imaged per condition in 14 independent experiments (total number of cells: $n_{GFP} = 121$, $n_{Src} = 73$). *$P < 0.05$ (Student's $t$-test). **c** Time-lapse imaging of Anillin-GFP during vSrc cell extrusion. Embryos were injected with the dUAS:myr-Cherry-vSrc;Anillin-GFP construct. Movies were taken over 4 h. Frames were extracted from a representative movie at indicated times where $t = 0$ is the moment of extrusion. White arrowheads indicate the position of the Anillin ring. Scale bars, 25 μm (*xy*) 10 μm (*xz*). **d** The effect of exogenous Anillin on rescuing the Anillin morphant phenotype cell-autonomously in vSrc-driven extrusion. Embryos were injected with the following constructs: dUAS:EGFP-vSrc and dUAS:EGFP-vSrc,Anillin-GFP and the Anillin morpholino. Data are mean ± s.d. of three independent experiments (total number of embryos: $n_{Src} = 33$; $n_{Src,Anillin} = 32$). *$P < 0.01$ (Student's $t$-test). **e** Quantification of the failed extrusion rate based on time-lapse imaging. Embryos were injected with the following constructs: dUAS:EGFP-vSrc;nucGFP and dUAS:EGFP-vSrc;DN-Anillin. Failed extrusion rates were calculated as the number of cells that rounded up and then returned to the monolayer without division or extrusion over 4 h by the initial number of GFP-positive cells per embryo. Each mark represents a division rate in a single embryo. Eleven embryos were imaged per condition in three independent experiments (total number of cells: $n_{Src} = 168$, $n_{Src,DNAnillin} = 149$). Grey lines represent mean values ± s.d. *$P < 0.05$ (Student's $t$-test). **f** Quantification of the angle between the Anillin ring and the surface of the embryo over time. Data from four cells acquired in three independent experiments were then aligned to the time of extrusion, $t = 0$, averaged ± s.d. and plotted

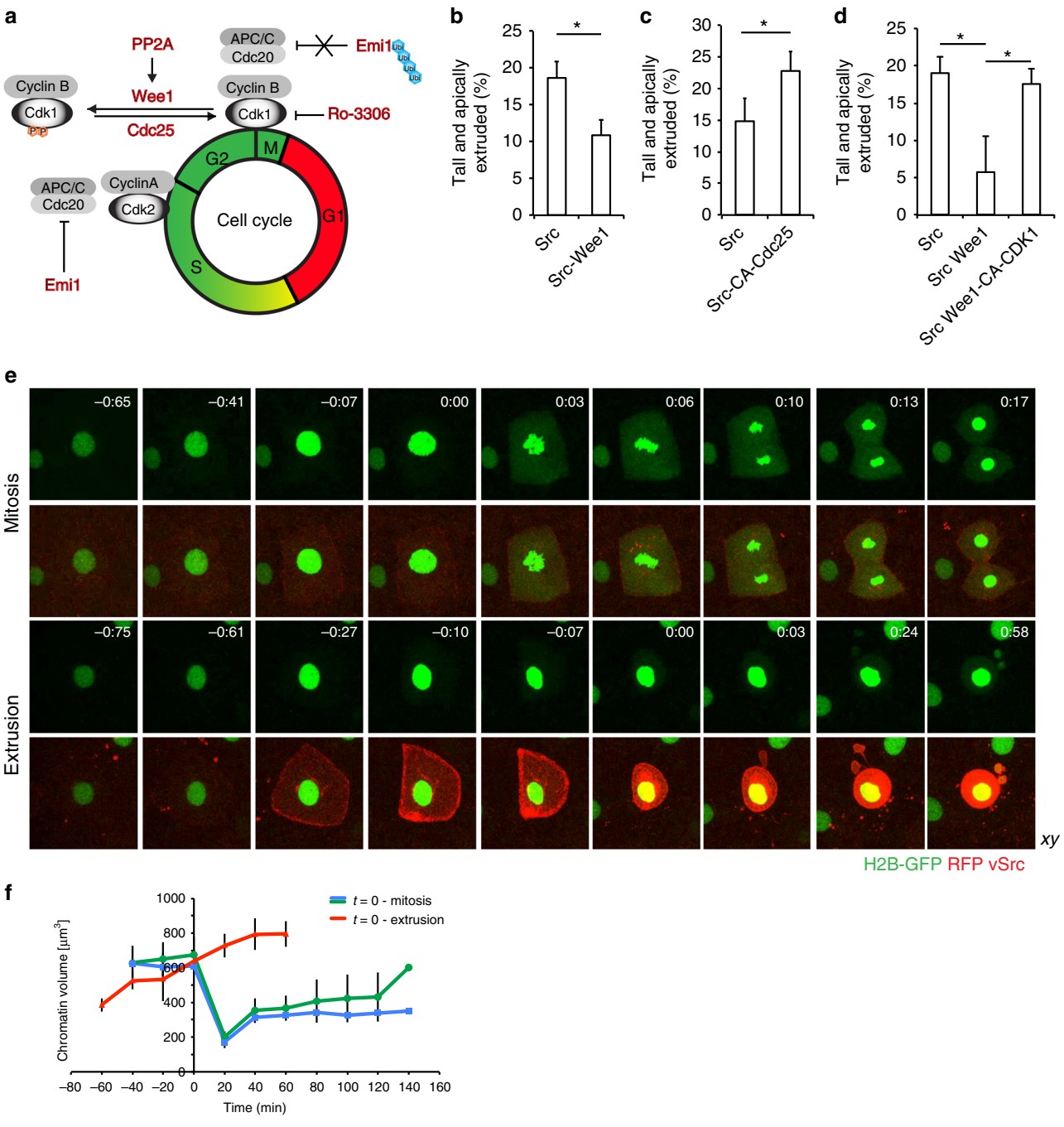

**Fig. 3** vSrc-transformed cells extrude in early mitosis. **a** A schematic model of cell cycle regulation. Highlighted in red are the molecules whose roles in cell extrusion were tested in this work. **b** The effect of Wee1 expression on vSrc-driven extrusion. Embryos were injected with the following constructs: dUAS: EGFP-vSrc and dUAS:EGFP-vSrc;Wee1. Data represent the number of tall and extruded cells divided by the total number of GFP-positive cells. Data are mean ± s.d. of three independent experiments (total number of embryos: $n_{Src}$ = 21; $n_{Src,Wee1}$ = 27). *$P$ < 0.05 (Student's $t$-test). **c** The effect of constitutively active Cdc25 on vSrc-driven extrusion. Embryos were injected with the following constructs: dUAS:EGFP-vSrc and dUAS:EGFP-vSrc;CA-Cdc25. Data are mean ± s.d. of three independent experiments (total number of embryos: $n_{Src}$ = 18; $n_{Src,Cdc25}$ = 21). *$P$ < 0.05 (Student's $t$-test). **d** Constitutively active Cdk1 rescues Wee1 inhibition of vSrc-driven extrusion. Embryos were coinjected with the following constructs: dUAS:EGFP-vSrc alongside either dKrt18:myr-Cherry, dKrt18:Cherry-Wee1 or dKrt18:Cherry-Wee1;CA-Cdk1. Data are mean ± s.d. of three independent experiments (total number of embryos: $n_{Src}$ = 30; $n$ = 29; $n_{Src,Wee1,Cdk1}$ = 33). *$P$ < 0.05 (Student's $t$-test). **e** Time-lapse imaging of H2B-GFP in mitosis (top panel) and extrusion (bottom panel). Embryos were injected with the dUAS:myr-Cherry-vSrc;H2B-GFP construct. Movies were taken over 4 h and frames were extracted from a representative movie. $t$ = 0 indicates either the beginning of mitosis (chromatin condensation) or the moment of extrusion. Scale bars, 10 μm. **f** Quantification of the chromatin volume (defined as the volume that H2B-GFP signal occupies in space) measured using the surface function of Imaris. The blue line follows chromatin volume change over time in a dividing EVL cell expressing myr-Cherry (averaged data from 3 cells), the green line—in a dividing vSrc cell (averaged data from 4 cells), the red line—in a extruding vSrc cell (averaged data from five cells). Error bars represent s.d.

CDK1 and allows mitotic entry. It is transcribed and stabilised in cells from the late S phase throughout the G2, and is rapidly degraded in mitosis[29]. Here we used it as a marker of the cell cycle phase in live-imaging as none of the previously established FUCCI markers[30–33] worked in our system. In the CcnB1-GFP line, the GFP signal was present in the cytoplasm and gradually increased before mitosis. Following G2/M transition, CcnB1-GFP localised to the nucleus[34] and was degraded during division (Fig. 4a, Supplementary Movie 7). To estimate the average length of different cell cycle phases, we acquired movies of CcnB1-GFP transgenic embryos. Some of the cells that divided at the beginning of each 8-hour movie divided again. For quantification, we split the cell cycle into three phases based on changes in the intensity and localisation of CcnB1-GFP: (1) mitosis, defined as the time from the nuclear import of Cyclin B1 until completed cytokinesis, (2) G1/S phase, the time from completed cytokinesis until the GFP signal reappeared in the cytoplasm, and (3) S/G2, the time of the GFP signal present in the cytoplasm until its nuclear import. The average length of the cell cycle was 8 h 25 min, with mitosis lasting 48 min and variable G1/S and S/G2 lengths of 3 h 14 min and 4 h 23 min, respectively (Fig. 4c).

To characterise how vSrc affects cell cycle parameters in the EVL, CcnB1-GFP transgenic embryos from the same batches were

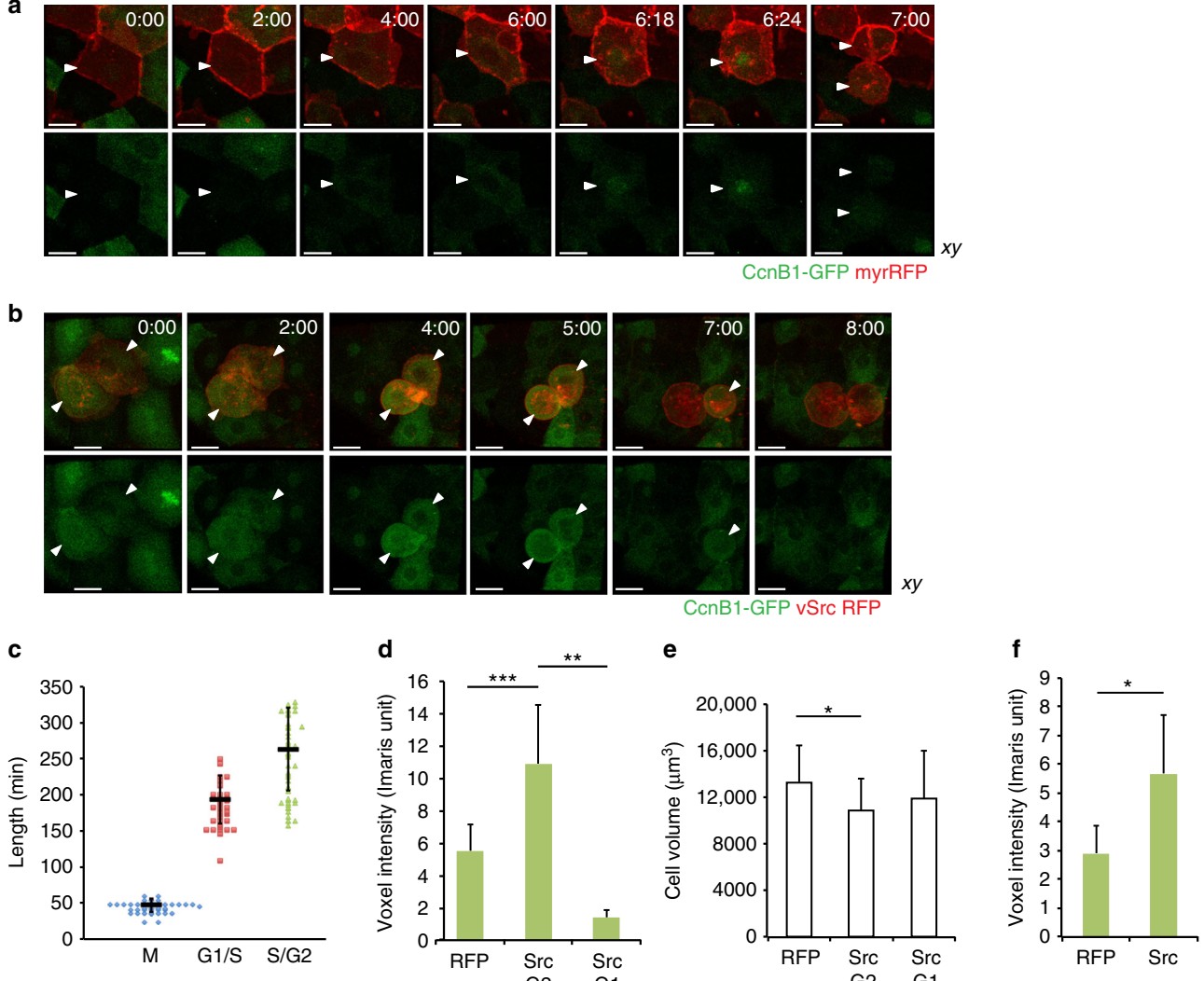

**Fig. 4** vSrc modulates cell cycle progression. **a**, **b** Time-lapse imaging of Cyclin B1-GFP in cell extrusion. Transgenic embryos obtained from a line expressing a cell cycle progression marker specifically in the EVL (Krt18:CcnB1-GFP) crossed with the Krt18:KalTA4-ERT2 line were injected with the construct UAS:myr-Cherry (**a**) or UAS:myr-Cherry-vSrc (**b**). Movies were taken over 8 h. Frames were extracted from a representative movie at indicated times from the tailbud stage. White arrowheads indicate cells that undergo mitosis or become extruded. Scale bars, 10 μm. **c** Length of different cell cycle phases of the EVL cells from the CcnB1-GFP line. The cell cycle phases are defined as follows; M is the time from the nuclear import of CcnB1-GFP until completed cytokinesis; G1/S is the time from completed cytokinesis until the GFP signal returns to the cytoplasm; S/G2/M is the time of the GFP signal present in the cytoplasm until mitosis. Each mark represents a single cell. Black lines represent mean values ± standard deviation (s.d.). Data collected from five movies in three independent experiments. **d** Mean voxel intensity ± s.d. of the CcnB1-GFP signal in single cells immediately before extrusion (myr-Cherry-vSrc) or division (myr-Cherry). Data collected from 31 extruded vSrc cells and 31 dividing myr-Cherry cells in 12 movies per condition from five independent experiments. ***$P = 2.1 \times 10^{-7}$ (Student's $t$-test), **$P = 8.4 \times 10^{-4}$ (Student's $t$-test). **e** Cell volume analysis before division and extrusion. Average volume ± s.d. of the cells used for CcnB1-GFP signal quantification immediately before extrusion or division (Fig. 4d). *$P < 0.05$ (Student's $t$-test). **f** Mean voxel intensity ± s.d. of the green channel (CcnB1-GFP) in myr-Cherry cells and myr-Cherry-vSrc cells at time 0 of a time-lapse from cells remaining in the epithelium. Data collected from 11 embryos per condition in five independent experiments. *$P = 0.005$ (Student's $t$-test)

injected with constructs carrying myr-Cherry-vSrc or myr-Cherry. Each embryo pair was then imaged and analysed together to avoid bias (Fig. 4a, b, Supplementary Movies 7 and 8). As a specific amount of the active Cyclin B1/CDK1 complex triggers mitosis[29], we measured GFP signal intensity prior to division or extrusion. Firstly, 28 out of 31 extruded vSrc cells in CcnB1-GFP embryos had a high GFP signal, indicating that extrusion occurred in the later phases of the cell cycle (Fig. 4b). The average GFP intensity in vSrc cells about to be extruded was nearly doubled in comparison to control cells before division (Fig. 4d). Three out of 31 vSrc cells were extruded soon after dividing within the EVL in the G1 phase, with low GFP intensity (Fig. 4d), indicating that a G1 extrusion can also occur, but is relatively rare. The average volume of vSrc cells before extrusion was significantly smaller than that of normal EVL cells before division, but this small decrease was not sufficient to explain the increased GFP intensity in vSrc cells (Fig. 4e). Moreover, the increased GFP signal was already observed in vSrc cells within the epithelium at the start of imaging ($t = 0$) as compared to control cells (Fig. 4f). Finally, when assessing how long vSrc cells remained in the S/G2 phase, we realised that the cells, which became extruded, were rarely GFP-negative (only in 2 out of 28 cases), making it impossible to measure the length of the S/G2 phase before extrusion. Together, these data indicate that vSrc cells before extrusion remain longer in the S/G2 phase and accumulate more Cyclin B1 than normal EVL cells prior to division. This suggests that G2/M arrest occurs before vSrc cell extrusion.

**Src-transformed mammalian cells extrude in two modes**. To elucidate whether coordination of oncogenic extrusion and the cell cycle was a general phenomenon, we took advantage of the previously established Madin–Darby Canine kidney (MDCK) mammalian tissue culture system[6]. Firstly, we treated mixed cultures of Src-expressing and normal MDCK cells (mixing ratio 1:50) with cell cycle inhibitors: a CDK1 inhibitor Ro-3306 (G2/M arrest; Supplementary Fig. 5A, B) and hydroxyurea (S phase arrest; Supplementary Fig. 5C). Consistent with the zebrafish data, we found that Src cell extrusion was inhibited by Ro-3306 (Fig. 5a). Further, to perform live-imaging of the cell cycle, we established a new line harbouring active Src and the cell cycle marker FUCCI[32]. FUCCI displays different nuclear signal depending on the phase of the cell cycle: red in G1 and green in S/G2. With this line, we found that in 50% of Src-transformed cells their nucleus turned red following extrusion (transition to G1 after extrusion; Fig. 5b and C). In some cases extrusion and division happened simultaneously, in others extrusion took place instead of mitosis (Fig. 5b). However, in 30% of Src-transformed MDCK cells their nuclei remained red throughout (G1 extrusion, Fig. 5c), implying a cell cycle-independent mode of extrusion, similar to the occasional events in the embryo. These data indicate that both the zebrafish embryo and mammalian cells utilise two modes of oncogenic extrusion, of which the more frequent one requires coordination with the cell cycle.

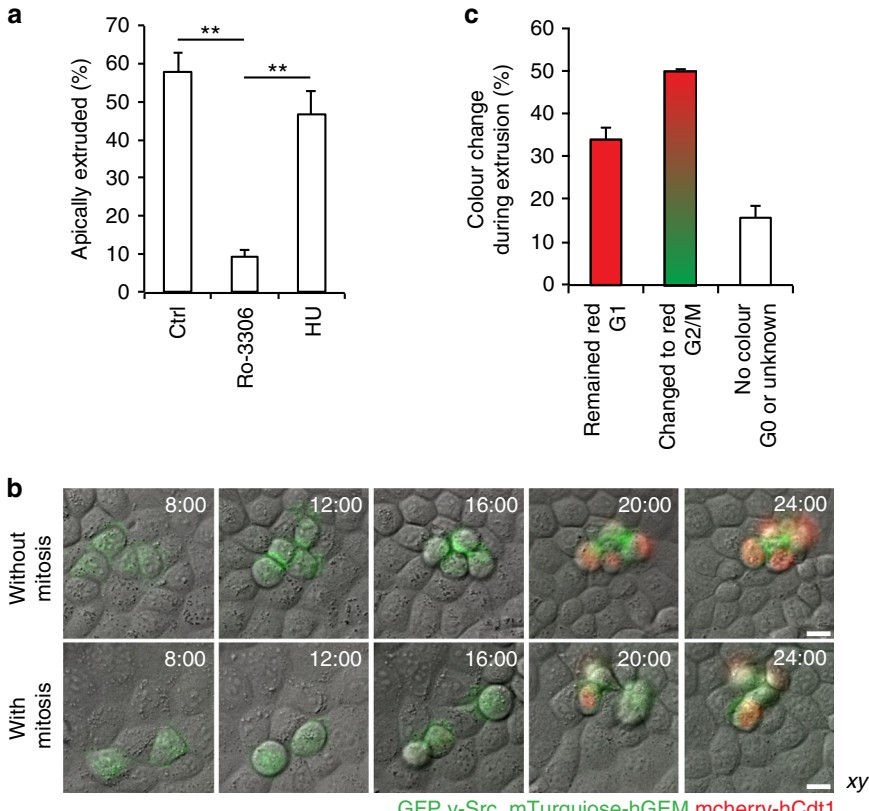

GFP v-Src   mTurquiose-hGEM   mcherry-hCdt1

**Fig. 5** Majority of Src-transformed MDCK cells extrude during or instead of mitosis. **a** The effect of inhibition of the G2/M transition or G1/S transition on extrusion of MDCK-pTR-cSrc-Y527F from a monolayer of normal MDCK cells. Inhibitors 2 mM hydroxyurea or 10 μM Ro-3306 were added with a 9-hour delay after the induction of Src-expression. At 24 h from Src induction cells were fixed, stained with phalloidin and imaged. Data are mean ± s.d. of three independent experiments (total number of cells: $n_{Ctrl} = 155$; $n_{HU} = 153$; $n_{Ro3306} = 152$). **P < 0.005 (Student's t-test). **b** Time-lapse imaging over 24 h of extrusion of MDCK stably expressing both pTR-cSrc-Y527F-GFP and FUCCI. GFP-CAAX labels Src cells (green membrane) that express Cherry-hCdt1 in G1 phase (red nuclei) or mTurquoise-hGEM in S/G2/M phases (green nuclei). **c** Quantification of the nuclear colour change in MDCK-pTR-cSrc-Y527F-GFP/FUCCI cells during extrusion based on live-imaging. Data are mean ± s.d. of two independent experiments (total number of extruded cells: n = 63)

**vSrc modulates cell cycle regulators Cdk1 and Pp2a.** To investigate how vSrc modulates the G2/M transition, we searched for Src-phosphorylated proteins in the database (www.phophositeplus.org), and speculated that two good candidates for mediating this process are CDK1 and PP2A. PP2A is a phosphatase that antagonises the mitotic CDK1-Cyclin B1 complex throughout the cell cycle, but must be inactivated in a cell entering mitosis[35]. Src phosphorylates PP2A on an inhibitory site tyrosine 307[36,37], promoting mitotic entry. As shown earlier, a mutant form of Pp2a lacking this phosphorylation, leads to inhibition of vSrc-induced extrusion (Supplementary Fig. 4A). Much less well documented is the action of the Src kinase on CDK1. The most probable Src-phosphorylation site within CDK1 is Tyrosine 15 (Y15), one of the two inhibitory sites that need to be dephosphorylated for activation of CDK1-Cyclin B1[38] (Fig. 6a). It has been demonstrated that Src phosphorylates a CDK1 peptide surrounding Y15 in vitro[37]. Therefore, we hypothesised that simultaneous phosphorylation of Pp2a and Cdk1 by vSrc, the former promoting, the later inhibiting G2/M transition, could be responsible for the prolonged G2 phase of vSrc cells. Hence, we tested if Cdk1 was a target of vSrc in our system.

Since an anti-CDK1-Y15 antibody did not recognise zebrafish Cdk1-Y15 at endogenous levels, we created the authentic CDK1 peptide[37] tagged with both RFP and an HA epitope. This peptide was phosphorylated soon before extrusion when expressed with GFP-vSrc, but not with GFP-CAAX (Fig. 6b). To consolidate this observation, we used the MDCK tissue culture system. Normal MDCK cells alone, Src-expressing MDCK cells alone and mixed cultures were analysed by western blotting with the anti-CDK1-Y15 antibody. A low level of phospho-CDK1 was observed in normal MDCK cells; the level was increased after incubation with hydroxyurea (G1/S arrest; Fig. 6c, d, original blots in Supplementary Fig. 6A, B). The phospho-CDK1 level was much higher in Src-expressing cells and moderate in the mixed cultures, suggesting that Src has the ability to directly or indirectly promote CDK1 phosphorylation and inhibit its activity (Fig. 6c, d).

A state in which cell cycle progression is both promoted and inhibited may result in "mitotic collapse"[39]. "Mitotic collapse" occurs after entry to mitosis, when CDK1 activation is not sustained at a level high enough for mitosis to proceed. This leads to dephosphorylation of mitotic substrates without degradation of Cyclin B1, resulting in cell death. Since Src activation both promotes and inhibits cell cycle progression, we wondered if cell extrusion could be a result of "mitotic collapse". To recapitulate this state in the EVL without vSrc, we simultaneously expressed two G2/M modulators: the inhibitor kinase Wee1 and the active phosphatase CA-Cdc25 both regulating CDK1. Although we managed to block EVL cells at the G2/M transition (as confirmed by nuclear localisation of Cyclin B1 in Supplementary Fig. 6C), no extrusion was observed over 4 h. Inflicting other mitotic defects such as triggering monopolar spindles by blocking Kif11 with an STLC inhibitor[40] also failed to cause extrusion (Supplementary Fig. 6D, E). We concluded that EVL cells cope with mitotic defects and delays without initiating death or extrusion of the "faulty cells" within the duration of our experiments.

Overall, Src activation leads to a G2/M arrest due to modulation of the cell cycle regulators Cdk1 and Pp2a. However, this modulation itself is insufficient to cause extrusion, suggesting that other vSrc effectors must be involved.

**vSrc modifies adherens junctions to recruit Anillin.** At this point, the means by which vSrc hijacked the cytokinetic machinery were still unclear. In a dividing cell, positioning of the mitotic plane is determined by the mitotic spindle[18]; however, the spindle was absent in vSrc cells undergoing extrusion (Supplementary Fig. 4D). This raised the possibility that the Anillin ring may be involved in extrusion via junctional constriction. Since RhoA activation promotes Anillin recruitment to the mitotic plane[17,19,20] and modulates junctional integrity[41], we investigated the effects of constitutively active and dominant negative RhoA on extrusion. Surprisingly, expression of either of these forms supressed vSrc-mediated rounding-up, but not extrusion (Fig. 7a).

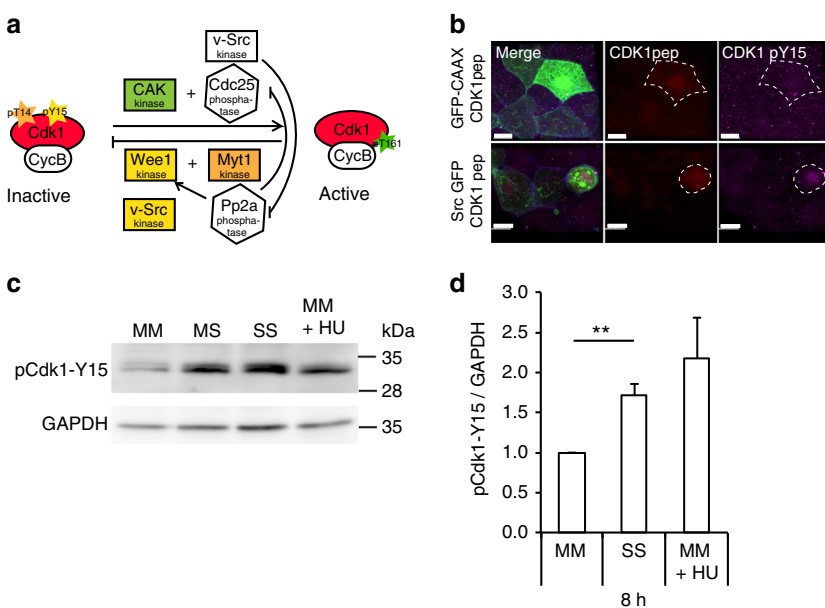

**Fig. 6** vSrc tyrosine kinase modulates cell cycle regulators CDK1 and Pp2a. **a** A schematic model of Src interference with cell cycle regulation. **b** Immunofluorescence images of CDK1 pY15 (purple) in the EVL cells expressing the mKO2-CDK1-pep (red) alongside EGFP or EGFP-vSrc (green). Scale bar, 10 μm. **c** The effect on phosphorylation of CDK1 after 8 h from Src activation in MDCK cells. MM MDCK cells alone, SS Src cells alone, MS cultures mixed 1:1, MM + HU MDCK cells treated with 2 mM hydroxyurea (HU). **d** Quantification of the mean normalised signal ± s.d. in western blotting with the anti-CDK1-pY15 antibody after 8 h from Src activation in MDCK cells from three independent experiments. **P < 0.005 (Student's t-test)

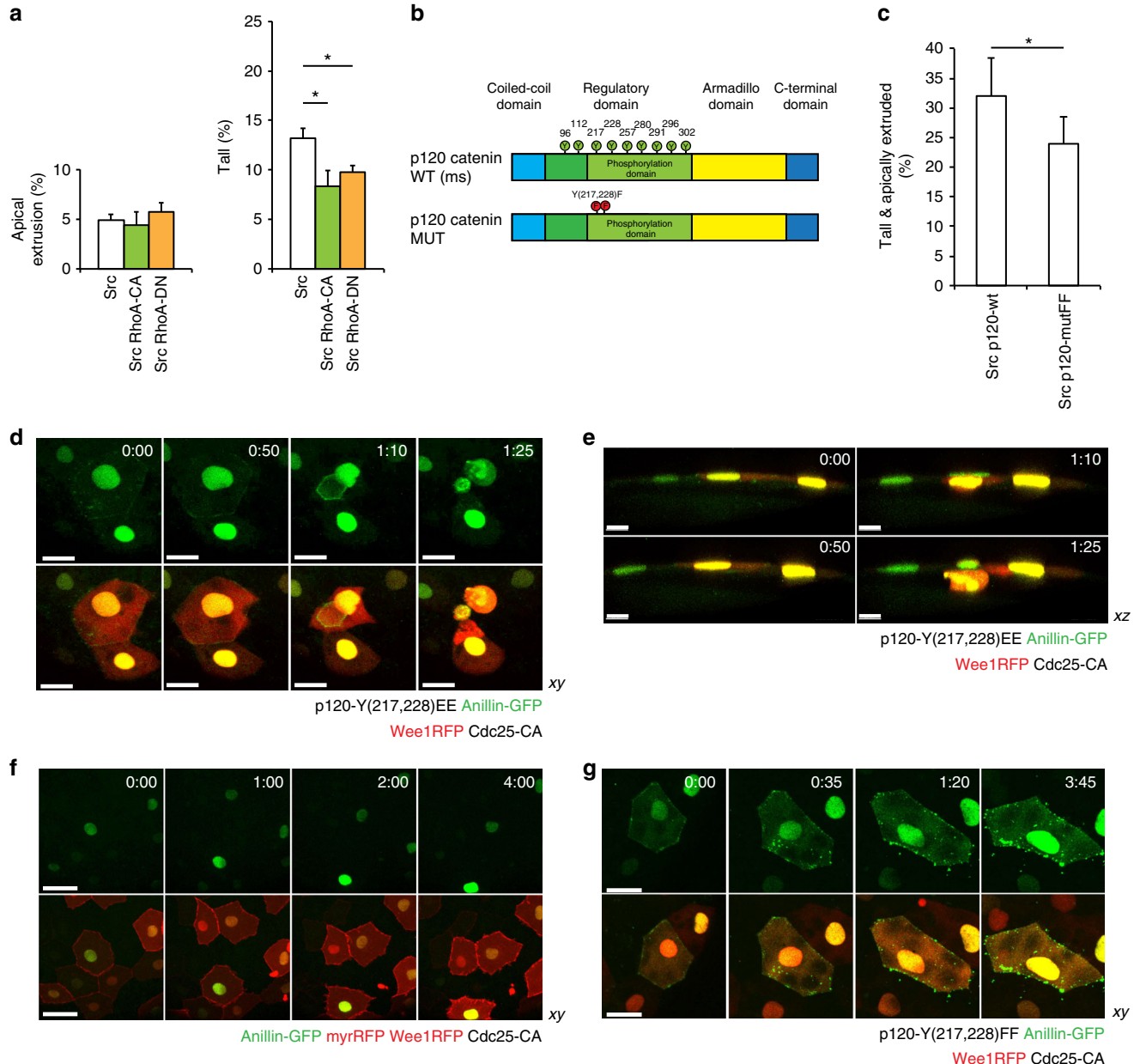

**Fig. 7** Src-phosphorylated p120-catenin recruits Anillin to the junctions. **a** The effect of constitutively active and dominant negative RhoA on vSrc-driven extrusion. Embryos were injected with the following constructs: dUAS:EGFP-vSrc, dUAS:EGFP-vSrc;CA-RhoA or dUAS:EGFP-vSrc;DN-RhoA. The two graphs display cells Apically extruded (outside of the embryo) and Tall (within the monolayer but taller than neighbours). Data are mean ± s.d. of 3 independent experiments (total number of embryos: $n_{Src} = 35$; $n_{Src,CA-RhoA} = 38$; $n_{Src,DN-RhoA} = 36$). *$P < 0.05$ (Student's $t$-test). **b** A schematic model of the domain composition of p120-catenin. In green are phosphorylation sites regulated by the Src kinase (from: PhosphoSitePlus database). In the bottom panel, a design of p120-catenin mutant with two sites regulating the interaction between p120-catenin and RhoA[44]. These sites are mutated from Y to F (p120-mutFF). **c** The effect of phosphomimetic p120-mutFF on vSrc-driven extrusion. Embryos were injected with the following constructs: dUAS:EGFP-vSrc;p120-wt or dUAS:EGFP-vSrc;p120-mutFF. Data are mean ± s.d. of three independent experiments (total number of embryos: $n_{Src,p120-wt} = 34$; $n_{Src,p120-mutFF} = 42$). *$P < 0.05$ (Student's $t$-test). **d**, **e** Time-lapse imaging of the effect of phosphomimetic p120-mutEE on the localisation of Anillin-GFP in cells arrested at the G2/M transition. Embryos were injected with a combination of the following constructs: dUAS:Cherry-Wee1;CA-Cdc25 and dUAS:p120-mutEE;Anillin-GFP. Movies were taken over 4 h. Frames were extracted from a representative movie at indicated times from the tailbud stage in $xy$ (**d**) and $xz$ (**e**) view. Scale bars, 40 μm (**d**) and 20 μm (**e**). **f** Time-lapse imaging of Anillin-GFP localisation in cells arrested at the G2/M transition. Embryos were injected with a combination of the following constructs: dUAS:Cherry-Wee1;CA-Cdc25 and dUAS:myr-Cherry;Anillin-GFP. Movies were taken over 4 h. Frames were extracted from a representative movie at indicated times from the tailbud stage. Scale bars, 50 μm. **g** Time-lapse imaging of the effect of p120-mutFF on the localisation of Anillin-GFP in cells arrested at the G2/M transition. Embryos were injected with a combination of the following constructs: dUAS:Cherry-Wee1;CA-Cdc25 and dUAS:p120-mutFF;Anillin-GFP. Movies were taken over 4 h. Frames were extracted from a representative movie at indicated times from the tailbud stage. Scale bars, 25 μm

Moreover, RhoA activation without vSrc did not trigger extrusion (Supplementary Fig. 7A). These results imply that focal RhoA activation at the junctions is necessary for the assembly of the contractile Anillin ring, but widespread modulation of RhoA presumably inhibits this process.

What mediates coupling of the cytokinetic machinery with the junctions? A study on the regulation of cytokinetic ring assembly identified p120-catenin, a component of the adherens junctions (AJs), as a scaffold that restricts RhoA activation to the constricting ring[42]. This prompted us to hypothesise that p120-catenin, a target of the Src kinase[43], could be the factor delocalising Anillin to the junctions in the absence of the mitotic spindle. To test this hypothesis, we modulated p120-catenin function in extruding vSrc cells. By phosphorylating the tyrosine residues Y217 and Y228 of p120-catenin, Src promotes the interaction between p120-catenin and RhoA[44]. Therefore, we created a phosphomimetic mutant version of p120-catenin in which these residues were replaced to mimic lack of phosphorylation (p120-mutant-FF, Fig. 7b). Expression of the p120-catenin-FF mutant together with vSrc significantly attenuated extrusion compared to the wild type version (Fig. 7c). This suggests that vSrc modulates AJs to couple them with the cytokinetic machinery.

We then sought whether modified p120-catenin alone could link the cytokinetic ring to the junctions in a cell cycle-dependent manner. Expression of the p120-catenin-EE mutant, mimicking a permanent state of phosphorylation, alongside Wee1 kinase and CA-Cdc25 phosphatase resulted in a G2/M arrest phenotype in normal EVL. In the presence of these three factors, we observed Anillin-GFP recruited to the junctions. Occasionally (2 cases in 5 movies), these cells underwent basal extrusion with immediate cell death (Fig. 7d, e, Supplementary Movie 9). Importantly, without p120-mutant-EE, expression of Wee1 and CA-Cdc25 was

not sufficient to recruit Anillin-GFP to the junctions (Fig. 7f). In rare cases of basal extrusion due to protein overexpression in these embryos, Anillin was not recruited to the junctions and this type of extrusion appeared to be Anillin ring-independent (Supplementary Fig. 7B). Finally, when p120-mutant-FF, instead of p120-mutant-EE, was expressed alongside Anillin-GFP in cells arrested at the G2/M transition, Anillin could not be stably recruited to the junctions, form a ring or facilitate cell extrusion (Fig. 7g, Supplementary Fig. 7C, D). This last observation proved that phosphorylation of p120-catenin by vSrc on residues Y217 and Y228 was necessary for the recruitment of Anillin to the junctions.

Collectively, Src activation in the EVL leads to assembly of a contractile ring through AJs in the prolonged G2 phase of the cell cycle and apicobasal extrusion via constriction of this ring in early mitosis.

**vSrc promotes apical polarity shift and survival.** So far, we found that vSrc must modify the cell cycle and AJs for extrusion. However, when we tried to mimic vSrc-like changes in cells without the active kinase, extrusion occurred only occasionally, was fully basal, and associated with cell death (Fig. 7d, e). We then wondered whether modulating cell polarity downstream of Src activation could lead to a change in directionality of extrusion. It has been shown that Src fine-tunes the activity of the small GTPase Cdc42 downstream of EGF stimulation[45]. Cdc42 has pivotal roles in establishing apicobasal polarity in all eukaryotic cells[46,47] and regulating the apical polarity complex Par3-Par6-aPKC in a manner conserved among different species[48,49] (Fig. 8a). Therefore, Cdc42 is a good candidate to link Src with polarity. When a dominant negative form of the Cdc42 mediator aPKC (DN-aPKC), consisting of the N-terminal regulatory

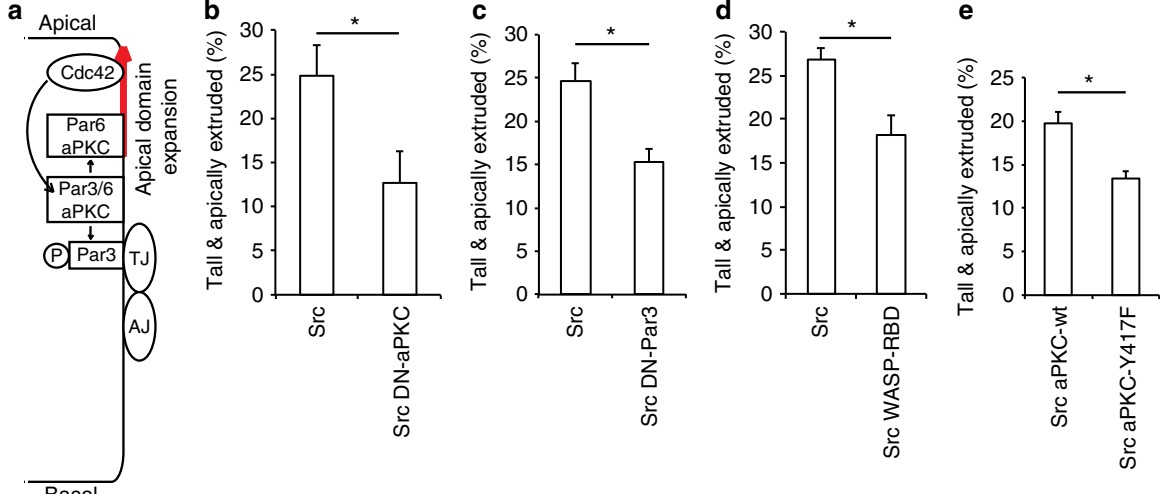

**Fig. 8** vSrc modulates apical polarity complex to facilitate extrusion. **a** A schematic model of regulation of apicobasal polarity by Cdc42 (adapted from ref.[49]). The Par3-Par6-aPKC complex is recruited to the cell surface in a Par3-dependent manner[46], and is regulated by apically localised Cdc42. Active Cdc42 stimulates phosphorylation, dissociation and relocation of Par3 to the apical-lateral border at tight junctions (TJ), while Par6-aPKC moves apically to define the apical domain[46,47,49]. **b** The effect of dominant negative aPKC on vSrc-driven extrusion. Embryos were injected with the following constructs: dUAS:EGFP-vSrc and dUAS:EGFP-vSrc;DN-aPKC. Data are mean ± s.d. of three independent experiments (total number of embryos: $n_{Src}$ = 31; $n_{Src,DN-aPKC}$ = 31). *$P$ < 0.05 (Student's $t$-test). **c** The effect of dominant negative Par3 on vSrc-driven extrusion. Embryos were injected with the following constructs: dUAS:EGFP-vSrc and dUAS:EGFP-vSrc;DN-Par3. Data are mean ± s.d. of three independent experiments (total number of embryos: $n_{Src}$ = 36; $n_{Src,DN-Par3}$ = 34). *$P$ < 0.01 (Student's $t$-test). **d** The effect of overexpression of WASP-RBD-GFP on vSrc-driven extrusion. Embryos were injected with the following constructs: dUAS:EGFP-vSrc and dUAS:EGFP-vSrc;WASP-RBD-GFP. Data are mean ± s.d. of three independent experiments (total number of embryos: $n_{Src}$ = 31; $n_{Src,WASP-RBD-GFP}$ = 36). *$P$ < 0.01 (Student's $t$-test). **e** The effect of phosphomimetic aPKC-Y417F on vSrc-driven extrusion. Embryos were injected with the following constructs: dUAS:EGFP-vSrc;aPKC-wt or dUAS:EGFP-vSrc;aPKC-Y417F. Data are mean ± s.d. of three independent experiments (total number of embryos: $n_{Src,aPKC-wt}$ = 39; $n_{Src,aPKC-Y417F}$ = 42). *$P$ < 0.05 (Student's $t$-test)

domain targeted to the membrane[50], was co-expressed with vSrc, it inhibited vSrc-driven extrusion (Fig. 8b). Similar result was achieved with a dominant negative version of Par3[51], another component of the apicobasal polarity pathway that delivers Par6-aPKC to the membrane (Fig. 8c), or expression of the Cdc42-binding domain WASP-RBD[52] that hinders Cdc42 activation (Fig. 8d). Since DN-aPKC had the most robust inhibitory effect on extrusion, we speculated that vSrc could also regulate aPKC independently of Cdc42. Further investigation uncovered a potential vSrc-phosphorylation site (determined using GPS 3.0 prediction server and literature[53,54]) on aPKC itself that was required for vSrc-driven extrusion (Fig. 8e). These results suggest the role for modulating apicobasal polarity in vSrc-mediated extrusion.

Apart from modulating the cell cycle, AJs and cell polarity, we speculated that vSrc promoted cell survival, as demonstrated previously[55]. Finally, to reconstitute vSrc-like cell extrusion, we expressed all the components: the cell cycle modulators Wee1 and CA-Cdc25, the AJs-mitotic plane coupler p120-mutant-EE, the polarity modulator active aPKC lacking the N-terminal regulatory domain (aPKC-delN) and the prosurvival protein XIAP in the EVL. Indeed, combining all the components mimicking Src activation resulted in vSrc-like extrusion: apicobasal split (Fig. 9a–c, Supplementary Movies 10 and 11) independent of cell death (Supplementary Fig. 8A). Thus, we managed to pinpoint four effector pathways downstream of vSrc that coordinate apicobasal extrusion: cell cycle, AJs, apicobasal polarity and cell survival (Fig. 9d, Supplementary Fig. 8B).

## Discussion

In this report, we have used the early zebrafish embryo to study oncogenic extrusion, primarily based on high-resolution live-imaging. We found that vSrc-mediated apicobasal extrusion is executed by hijacking the cell cycle and rewiring cytokinesis. vSrc drives EVL cells into the G2 phase and initially blocks further progress into mitosis. During this period, Src activation leads to the reorganisation of the zonula adherens (ZA) through incorporation of Anillin recruited to AJs by vSrc-modified p120-catenin. With a contractile junctional ring assembled, the cell enters mitosis and the ring constricts, facilitating an apicobasal split of the vSrc cell. During extrusion, the larger part of the cell containing the nucleus is released apically. Thus, premature and rewired cytokinesis occurs in early prophase before NEB or mitotic spindle assembly.

Such premature cytokinesis has been described in unfertilised syncytial eggs in *Drosophila* physiologically remaining in the M-phase[56], despite the fact that there are no microtubule bundles present in these eggs outside of a small peripheral meiotic spindle. When injected locally with CDK1 inhibitors or active–RhoA, the embryo forms de novo a premature contractile structure that resembles a cytokinetic ring with Actin, Myosin and Anillin at the

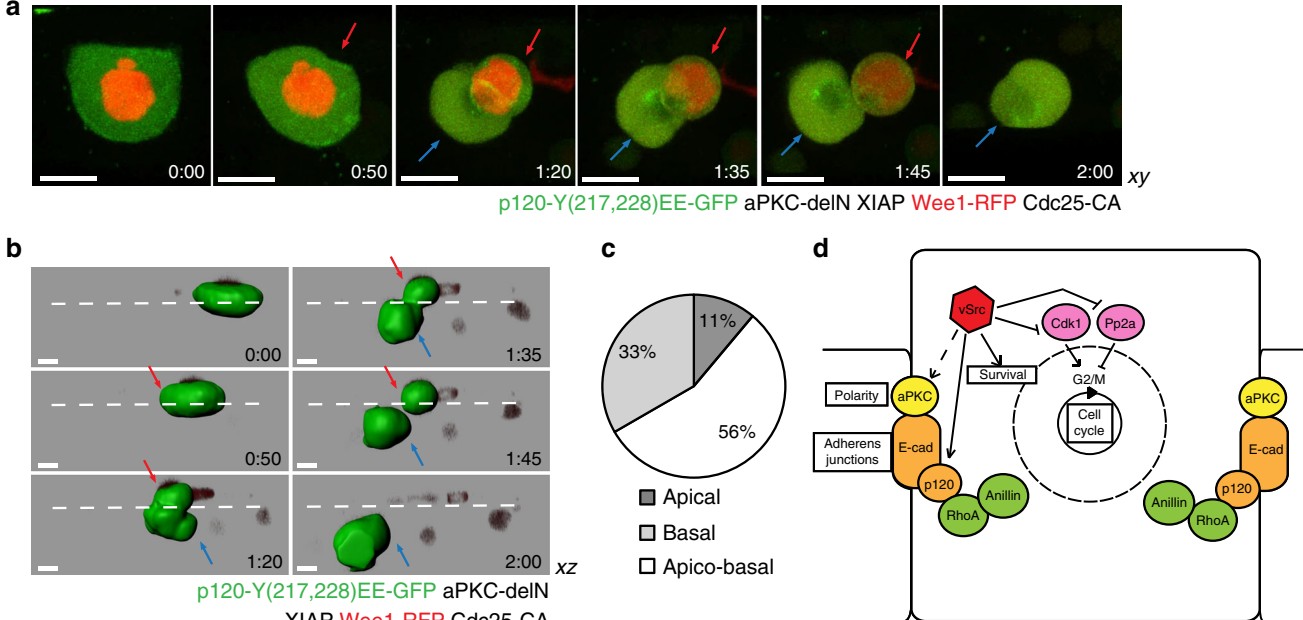

**Fig. 9** Modulating the cell cycle, junctions and polarity mimics vSrc-like extrusion. **a** Time-lapse imaging of vSrc-like extrusion induced by coexpression of p120-mutEE, aPKC-delN and the apoptotic inhibitor XIAP in G2/M-arrested cells. Embryos were injected with a combination of the following constructs: dUAS:Cherry-Wee1;CA-Cdc25, dUAS:p120-mutFF;aPKC-delN and Krt18:XIAP. Movies were taken over 6 h. Frames were extracted from a representative movie at indicated times from the tailbud stage. The cell is undergoing an apicobasal split (apical part is marked with red arrows and the basal part with blue arrows). Scale bars, 15 µm. **b** Time-lapse imaging of vSrc-like cell extrusion in **a** segmented using the Imaris software. The surface function was used to segment GFP-positive cells over time. In this cross section of the embryo (*xz* view), the cell is undergoing an apicobasal split (apical part is marked with red arrows and the basal part with blue arrows). The dashed white line indicates the surface of the embryo. Scale bars, 10 µm. **c** Quantification of the type of vSrc-like cell extrusion based on time-lapse imaging. Extrusion was induced by coexpression of p120-mutEE, aPKC-delN and the apoptotic inhibitor XIAP in G2/M-arrested cells. Fourteen embryos were imaged in four independent experiments (total number of extrusions: $n_{p120EE,aPKC-delN,wee1,cdc25,XIAP} = 27$). **d** A schematic model of vSrc-driven cell extrusion. vSrc interferes with the cell cycle, and modulates adherens junctions, cell survival and apicobasal polarity, leading to apicobasal extrusion. vSrc-expressing cell becomes taller than its neighbours. Cell cycle regulators are hijacked; Pp2A is inactivated earlier in the cell cycle, but counteracting Cdk1 inhibition results in G2/M arrest instead of mitosis. The nuclear envelope becomes partially permeable and Anillin is recruited to the adherens junctions by modified p120-catenin, presumably through active RhoA. A contractile junctional ring assembles parallel to the plane of the epithelium, constricts in early mitosis and releases the cell from the epithelium. vSrc-mediated modulation of the apicobasal polarity complex (e.g. aPKC) promotes the predominantly apical direction of extrusion. Immediate cell death is avoided due to vSrc promoting cell survival

site of injection. In our system, without a mitotic spindle, vSrc appears to generate a narrow zone of active RhoA by modulating p120-catenin[42–44]. This modulation results in recruitment of the cytokinetic scaffold Anillin and assembly of a premature mis-oriented cytokinetic ring by rebuilding the existing junctional Actomyosin ring. To resolve the dual roles of RhoA in mediating cytokinesis and junctional integrity in extruding vSrc cells, optogenetic approaches will be necessary in the future[57].

Transformation of a junctional Actomyosin ring into a contractile ring has been described previously in the context of apoptotic extrusion, where the ring was formed in the neighbours[58]. The process relies on the reorganisation of short into co-aligned peri-junctional Actin bundles mediated by Coronin B1. When a cell dies, two junctional Actomyosin rings can be seen at the level of AJs: one in the dying cell and the other in the neighbours. The neighbouring ring becomes thicker and relocalises basally to facilitate apical extrusion, while the ring in the dying cell remains stationary. In contrast, during vSrc cell extrusion, the autonomous ring becomes contractile and relocalises from the junctions to an oblique position. Despite being fundamentally different, either autonomous or non-autonomous, these two processes are also remarkably similar. Both rely on modification of the ZA into a contractile ring to facilitate extrusion and appear regulated by RhoA[59]. Although the Actomyosin ring in neighbours of a dying cell is controlled by p115RhoGEF[60], potential regulators of the autonomous contractile ring remain to be identified in the future.

Another issue to consider is the non-cell-autonomous contribution to apoptotic and oncogenic cell extrusion. Michael et al.[58] show that assembly of a contractile ring in neighbouring cells depends on apoptotic shrinkage of the dying cell, pulling on the neighbouring junctions and mechanosensing through E-cadherin. In case of vSrc-induced extrusion, a cell-autonomous active constriction via the pseudo-cytokinetic ring generates a force that presumably pulls on the AJs of the neighbours. Although the direction of the forces that act on neighbouring AJs in dead and oncogenic extrusion is the same, the strength would likely differ. Together, these findings raise the intriguing possibility that the pulling force on E-cadherin determines the mechanism used for extrusion by the neighbouring cells. Importantly, there is a difference in the non-cell-autonomous response. The Actin ring in the neighbours is less pronounced in Src-driven extrusion than in apoptotic extrusion[6,59]. Instead, neighbours of transformed cells employ the Actin cross-liking protein Filamin to facilitate this process[2]. Remarkably, non-autonomous recruitment of Filamin is also regulated by RhoA[4,59]. However, different RhoA GEFs/GAPs could be involved in the regulation of RhoA and Filamin in extrusion of transformed cells. In future studies, it will be crucial to clarify whether E-cadherin signalling acting as a mechanosensor could be upstream of differential RhoA activation in the neighbours and to separate RhoA regulators recruiting Actin or Filamin in each of these processes.

It appears that RhoA GEFs/GAPs in transformed cells and their neighbours are also key in controlling the direction of extrusion. p115RhoGEF has a crucial role in determining where the Actomyosin ring is assembled in the neighbours of a dying cell[60]. Moreover, a recent study of oncogenic extrusion in the *Drosophila* wing imaginal disk implicates RhoGEF2, a fly homologue of p115RhoGEF, in determining directionality[61]. The presence of RhoGEF2 is linked to "tumour hotspots" with predominant apical extrusion whereas "tumour coldspots" are associated mostly with basal extrusion. However, our results (Fig. 9a–c) show that autonomous polarity change is sufficient to reverse the direction of extrusion. These seemingly opposing observations on controlling directionality of extrusion cell-autonomously or non-cell-autonomously may be consolidated

by the hypothesis that positioning of the ZA between transformed cells, and their neighbours is responsible for driving extrusion[58]. Since AJs give rise to the Actomyosin rings facilitating extrusion, their positioning should be crucial for the direction of extrusion and could be regulated from both sides: the extruding cell and its neighbours.

Our data reveal that autonomous regulation of apicobasal polarity is necessary for vSrc-mediated extrusion (Fig. 8b–e) and contributes to regulating its directionality. Although precise determination of how the apical polarity complex is controlled in extrusion is beyond the scope of this paper, we believe that both Cdc42 and aPKC could be activated by vSrc. While the Cdc42 phosphorylation site was already established[45], aPKC regulation was demonstrated only indirectly[53,54]. In the future it will need to be clarified whether potential direct activation of aPKC is involved. Recent findings on apical domain expansion in epithelial cells shed light on how this process could be involved in extrusion[49,62]. Dbl3 is a regulatory GEF of Cdc42, a GTPase necessary for oncogenic extusion[63]. Dbl3 directs apical localisation and activation of Cdc42 and expansion of the apical domain through the regulation of the apical polarity complex aPKC-Par3-Par6. Downstream of Cdc42, the myosin kinase MRCK promotes myosin flow that separates apical aPKC-Par6 from junctional Par3, a step crucial for epithelial differentiation. Interestingly, MRCK was found differentially phosphorylated in H-Ras$^{V12}$ cells interacting with normal cells[5]. Moreover, overexpression of Dbl3 promotes apical expansion resembling rounding-up before extrusion[62]. Hence, this pathway may promote rounding alongside RhoA in cell cycle-dependent extrusion or on its own in cell cycle-independent extrusion. Further studies will be necessary to clarify this point.

Finally, it will be worth investigating whether cell extrusion induced by other oncogenes occurs in a cell cycle-dependent manner. If the mechanism is similar, blocking proliferation to treat carcinogenesis may impair the primary EDAC response and should be reconsidered.

Overall, our study uncovers a novel mechanism underlying EDAC. Further investigation will lead to identifying regulators of GTPases (in particular, RhoA) that control different aspects of extrusion in both cell-autonomous and non-cell-autonomous mechanisms. Understanding the coordination of timing, apical polarity and junctional integrity may eventually result in potential therapies to boost EDAC.

## Methods

**Constructs**. All the constructs used for experiments were based on the pBR-Tol2 vector with either Krt18 promoter or the 5xUAS element-driving expression in one (UAS, Krt18) or both (dUAS, dKrt18) directions[2,6,8]. To generate the dKrt18 vector, we placed the endogenous basal promoter (−150 bp from the transcription staring site of *krt18* gene) to −5 kb upstream of the EVL-regulatory sequence of the Krt18 promoter in the reverse complementary orientation (Fig S1). Previously published constructs used in this work were: dUAS:EGFP-vSrc[8], UAS:EGFP-vSrc[2,8] and UAS:myr-Cherry-vSrc[2]. On the basis of the four basic pBR-Tol2 vectors with single or double promoters, we created a number of new constructs using the InFusion system (Clontech). To make new constructs in most cases, we used cDNA from zebrafish embryos unless otherwise indicated. The following previously cloned cDNAs were gifts: Wee1 from David Whitmore[64], CA-Cdc25 (Cdc25-3S/T-A) from David Kimelman[30], aPKC (rat) from Sergei Sokol[65], Dcx-GFP from Marina Mione[66], H2B-GFP and Par3-GFP from Jon Clarke, Anillin from Luccia Poggi[22], p120-wt (mouse) from Roberto Mayor[67], DAPK1 from Caroline Brennan and WASP-RBD-GFP from Karl Matter. The following cDNAs were cloned from a cDNA library created using 24-h-old zebrafish embryos: Pp2a (ZDB-GENE-050417-441), RhoA (ZDB-GENE-040426-2150), CcnB1 (ZDB-GENE-000406-10), Cdk1 (ZDB-GENE-010320-1) and XIAP (ZDB-GENE-030825-7) based on the ZFIN database. Indicated point mutations and deletions were achieved using the InFusion method (Clontech) and confirmed by sequencing (Source BioScience). The specific created mutations were: Y307F in CA-Pp2a, Q63L in CA-RhoA, T19N in DN-RhoA, T14A and Y15F in CA-Cdk1, Δ(1-740) in DN-Anillin, Δ(201-591) in DN-aPKC, Y417F in aPKC-mutF, Δ(1-201) in aPKC-delN, Y217,228 F in p120-mutFF, Y217,228E in p120-mutEE and Δ(375-1127) in DN-ParD3. nucGFP was

created by fusing 2xNLS Sv40 with NLS from Wee1 (RNNNRKRSHWN), hmAzami-Green and EGFP. CDK1pep-expressing construct was created by fusing FLAG, mKO2 fluorophore, CDK1 peptide (KIEKIGEGTYGVVYK) and 2xHA tag.

On the basis of the dUAS:EGFP-vSrc construct, we created: dUAS:EGFP-vSrc; Wee1, dUAS:EGFP-vSrc;CA-Cdc25, dUAS:EGFP-vSrc;p20, dUAS:EGFP-vSrc;p21, dUAS:EGFP-vSrc;Pp2a-Y307F, dUAS:EGFP-vSrc;CA-RhoA, dUAS:EGFP-vSrc; DN-RhoA and dUAS:EGFP-vSrc;DN-aPKC. We also replaced the EGFP in the dUAS:EGFP-vSrc construct with myr-Cherry to obtain dUAS:myr-Cherry-vSrc and subsequently used it to make the following constructs: dUAS:myr-Cherry-vSrc; XIAP, dUAS:myr-Cherry-vSrc;Dcx-GFP, dUAS:myr-Cherry-vSrc;H2B-GFP, dUAS:myr-Cherry-vSrc;nucGFP, dUAS:myr-Cherry-vSrc;Anillin-GFP, dUAS:myr-Cherry-vSrc;DN-Anillin-GFP, dUAS:myr-Cherry-vSrc;p120-wt and dUAS:myr-Cherry-vSrc;p120-mutFF. We used the original pBR-Tol2-dUAS vector to create the following constructs: dUAS:Cherry-Wee1;CA-Cdc25, dUAS:p120-mutEE; Anillin-GFP, dUAS:myr-Cherry;Anillin-GFP, dUAS:p120-mutFF;Anillin-GFP, dUAS:p120-mutFF;myr-Cherry, dUAS:myr-Cherry;GFP-Emerin, dUAS:Dcx-GFP; H2B-RFP, dUAS:myr-Cherry;H2B-GFP, dUAS:myr-Cherry;Anillin-GFP, dUAS: GFP-CAAX;CA-RhoA and dUAS:myr-Cherry;DAPK1. We used the pBR-Tol2-UAS to create: UAS:myr-Cherry. We used pBR-Tol2-Krt18 to create the following constructs: Krt18:XIAP, Krt18:CcnB1-GFP. We used the pBR-Tol2-dKrt18 to create the following constructs: dKrt18:H2B-GFP;myr-Cherry, dKrt18:myr-Cherry, dKrt18:Cherry-Wee1 and dKrt18:Cherry-Wee1;CA-Cdk1.

**Antibodies, morpholinos and inhibitors.** Anti-GFP antibody was from Abcam (13970), 1:1000. Anti-RFP antibody was from MBL (PM005), 1:1000. Anti-phospho-CDK1 Tyr15 antibody was from Cell Signaling (4539) used at 1:50. Anti-phospho-MLC2 Thr18/Ser19 antibody was from Cell Signaling (3674), 1:50. Anti-phospho-Histone H3 Ser10 antibody was from Upstate (MERCK: 06-570), 1:500. Anti-active Caspase 3 antibody was from BD Biosciences (559565), 1:300. Secondary antibodies were from Invitrogen Molecular Probes, 1:300. Phalloidin-Atto 647 N was from Sigma, 1:1000.

For knockdown experiments in zebrafish, we used Emi1 MO, a gift from Jon Clarke and Anillin MO[22] purchased from Gene Tools. 1 nL of 0.5 mM morpholino (Emi1 MO, control MO or Anillon MO) solution was injected into the yolk following a DNA injection. To generate mitotic spindle defects, Eg5 (Kif11) inhibitor STLC from Alfa Aesar was used at 0.874 mM added together with tamoxifen, at 50–70% epiboly and during imaging.

**Generation and maintenance of transgenic fish lines.** The maintenance of fish and the collection of embryos were performed according to the internationally recognized guidelines[68]. Ethical approval for zebrafish experiments was obtained from the Home Office UK under the Animal Scientific Procedures Act 1986. The line Tg(Krt18:KalTA4-ERT2) was previously established[2,8]. To establish Tg(Krt18: Lifeact-Ruby) and Tg(Krt18:CcnB1-GFP) lines, we used the vector pBR-Tol2-Krt18 generated previously[6], and transferred Lifeact-Ruby[69] and CcnB1-GFP (see Constructs), respectively, downstream of the Krt18 promoter into the PmeI site. The resulting constructs (30 pg) were then coinjected with Tol2 RNA (7.5 pg) in the morpholino buffer (5 mM HEPES pH 7.5, 200 mM KCl) into one-cell wild-type embryos. The embryos positive for RFP (Lifeact-Ruby) and GFP (CcnB1-GFP) expression at 10 h post fertilization (hpf) were raised to adulthood, and crossed with wild-type fish to identify founder fish. Embryos from potential founders were imaged to select the optimal level of expression at which no overexpression phenotype could be observed. The founder fish were out-crossed with WT, and the F1 fish were selected on the basis of their fluorescent signal. All the embryos for experiments were obtained from crossing fish heterozygous with the Tg(Krt18: KalTA4-ERT2) line.

**Microinjection and confocal imaging of zebrafish embryos.** Embryos were injected with a single construct (16–20 pg) or multiple constructs (combined amount of DNA was 20 pg) and Tol2 RNA in the morpholino buffer (5 mM HEPES pH 7.5, 200 mM KCl) into the cell at one-cell stage, and treated with 0.5 mM 4-hydroxy tamoxifen (Sigma H7904, a stock of 5 mM in ethanol) at 50–70% epiboly[2]. For live-imaging, after 2 h of treatment, embryos were mounted in 0.8% low-melting agarose in fish water prior to confocal analysis. For immuno-fluorescence and quantification of extrusion rates, embryos were fixed in 4% PFA/ PBS at 2.5–3 h after induction, stained and mounted in 1% low-melting agarose in PBS prior to confocal analyses. Movies were taken over 4 h or over 8 h (cell cycle analysis with the Krt18:CcnB1-GFP line). Confocal images were taken using a ×25 0.95 NA water-immersion lens on a high-resolution single photon microscope Leica TCS SP8 and were analysed using the Imaris software (Bitplane).

**Immunostaining of fish embryos.** At 10 hpf GFP-positive or RFP-positive embryos were selected and dechorionated in 1% agarose plates to avoid damage. Embryos were fixed in a fresh solution of 4% PFA/PBS overnight at 4 °C and subsequently washed 3× for 10 min in PBS. Permeabilisation was performed for 15 min in PBS/0.5%TritonX-100 (PBSTr). Blocking in PBSTr/10% Goat serum/1% DMSO (Blocking buffer) lasted >1 h. Embryos were incubated with 1st antibody in 200 μl in Blocking buffer @ 4 °C overnight, then washed with PBSTr 3–6× for 30 min in total. Incubation with 2nd antibody in 200 μl Blocking buffer lasted 3–4 h at

room temperature, followed by washes with PBSTr 3–6× for 30 min in total. Phalloidin staining was performed for 30 min in PBSTr/10% Goat serum. For the phospho-CDK1 staining in the embryo, we expressed a CDK-consensus peptide tagged with RFP and HA alongside vSrc. We stained for phospho-CDK1 with an antibody. The shape of the GFP and vSrc cells was outlined in the Fig. 6b to make the distinction between the non-specific and specific signals clearer.

**Cell culture experiments.** MDCK cell lines were used in this study. The parental MDCK cells were a gift from Walter Birchmeier. MDCK and MDCK-pTR-cSrc-Y527F-GFP lines were cultured in DMEM containing 10% FCS (tetracycline free), penicillin/streptomycin, 5 μg ml$^{-1}$ of blasticidin (Invivogen) and 400 μg ml$^{-1}$ of zeocin (Invivogen)[2,6,63]. To establish MDCK-pTR-cSrc-Y527F-GFP line stably expressing FUCCI cell cycle markers mCherry-hCdt1(30/120) and mTurquoise-hGem(1/110), MDCK-pTR-cSrc-Y527F-GFP cells were transfected with P2A Fucci2.2_pCSII-CMV vector (a kind gift from Dr. Miyawaki) together with a pcDNA3.1 as a selection vector using Lipofectamine 2000 (Life Technologies), followed by selection in the medium containing 800 μg ml$^{-1}$ of G418 (Gibco), 5 μg ml$^{-1}$ of blasticidin and 400 μg ml$^{-1}$ of zeocin. To induce Src-expression, 2 μg ml$^{-1}$ of tetracycline (Sigma-Aldrich) was added to the medium. For immuno-fluorescence and time-lapse experiments, cells were cultured on type-I collagen gels from Nitta Gelatin (Nitta Cellmatrix type 1-A; Osaka, Japan)[6] neutralized on ice to a final concentration of 2 mg ml$^{-1}$ according to the manufacturer's instructions. For immunofluorescence, mixed cultures of cells (MDCK: Src = 50: 1) were plated and incubated for 8 h, before adding tetracycline. To avoid differences in cell density which could affect extrusion rates, proliferation inhibitors were added to the medium 16 h after tetracycline at following concentrations: hydroxyurea (2 mM) or Ro-3306 (10 μM). Cells were fixed and stained according to standard protocols[5]. Immunofluorescence images were taken with the Olympus FV1000 or FV1200 system and Olympus FV10-ASW software. Images were analysed with MetaMorph software (Universal imaging). For time-lapse imaging, following a 4 hour-tetracycline treatment, small groups of GFP-positive cells were chosen for imaging with Olympus IX81-ZDC (Olympus) and images were taken and analysed with Metamorph software (Molecular Devices). For Western blotting cells were plated in plastic dishes and induced with tetracycline for 8 h before lysis. Western blotting was carried out according to standard protocols[70]. Primary antibodies were used at 1:1000. The western blotting data were analysed using ImageJ (NIH). Original uncropped blots were included in Supplementary Figures. For FACS analysis, MDCK cells were incubated with or without proliferation inhibitors as before for 9 h prior to staining with Hoechst 33342 dye (1 mg ml$^{-1}$; ThermoFisher Scientific). After trypsinisation and straining, cells were counted, resuspended in 2% FBS/PBS, stained with propidium iodide and analysed for DNA content using FACSAriaTM II (BD Biosciences).

**Data analysis.** For data analyses, two-tailed Student's t-tests were used to determine P-values. P-values <0.05 were considered to be significant. Extrusion rates in fixed embryos were expressed as the number of extruded and tall cells (unless indicated otherwise) by the total number of GFP-positive or RFP-positive cells in the embryo. As extruded we classified cells that are no longer a part of the monolayer (their junctions closed-off or nearly closed off up to 90%). As tall we classified cells that were at least double the height of an average EVL cell, displaying signs of early extrusion, such as rounding. Only embryos with between 5 and 50 GFP-positive or RFP-positive cells were taken into consideration.

Proliferation rates in living embryos were expressed as the number of divisions over 4 h by the total number of cells at the beginning of the movie. Only embryos with between 5 and 35 GFP-positive or RFP-positive cells were taken into consideration. To measure chromatin volume, H2B-GFP signal was used to segment the GFP-positive region in the cell undergoing division or extrusion over time using the surface function of Imaris software. A constant threshold was used to avoid bias between different movies. The moment of mitosis or extrusion was set as point 0 and volumes from different movies were aligned according to time before and after extrusion and averaged to create graph Fig. 3f. To measure the intensity of the CcnB1-GFP signal, segmentation was performed in the red channel on the basis of the signal from the cell surface marker myr-Cherry using the Surface function of the Imaris software. The segmented cell surface was then used to calculate the average intensity of the green channel and cell volume (Fig. 4d–f). To define the position of the Anillin ring, Imaris spot function was used to determine points within the plane of the ring and the plane of the surface of the embryo. Extracted coordinates of the spots where then fed into a MATLab function (based on affine_fit(X)) to calculate the angle between two planes (Fig. 2f). Quantification of Anillin recruitment to the junctions by either p120-mutEE or p120-mutFF, was performed using automatic detection of Anillin spots with the Spots function of the Imaris software. Spot intensity per cell was then averaged for each of the three time points. Position of the spots in regards to the cortex (away or within the cortex; Supplementary Fig. 7C) was also assessed (Supplementary Fig. 7D).

**Code availability.** MatLab code used to quantify the angle of the Anillin ring can be found under https://github.com/ucbtkaw/Src-transformed-cells-hijack-mitosis-to-extrude-from-the-epithelium.git

## Data availability

The datasets generated and/or analysed during the current study are available from the corresponding author on reasonable request.

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

## Acknowledgements

We thank members of the Wilson, Rihel, Bianco and Tada labs for help, advice and sharing reagents, and the UCL fish facility for excellent zebrafish care. We would like to thank David Whitmore, David Kimelman, Sergei Sokol, Marina Mione, Jon Clarke, Luccia Poggi, Roberto Mayor, Caroline Brennan and Karl Matter for constructs they provided. We especially would like to thank Buzz Baum, Karl Matter, Jon Clarke, Richard Poole and Snezhka Oliferenko for critical reading of the manuscript and advice. This work is funded by the Cancer Research UK, A15936. Y.F. was supported by Japan Society for the Promotion of Science (JSPS) Grant-in-Aid for Scientific Research on Innovative Areas 26114001, Grant-in-Aid for Scientific Research (A) 26250026, the Naito Foundation and the Takeda Science Foundation.

## Author contributions

K.A. designed experiments, generated and analysed most of the data. M.K. generated MDCK-FUCCI line. K.A., M.K. and R.N. performed live-imaging in MDCK cells. R.N. performed western blots in MDCK cells. M.T. conceived and designed the fish model system and generated most fish lines. Y.F. conceived and designed the MDCK model system. K.A. and M.T. conceived and designed the study. K.A. created the figures. The manuscript was written by K.A. and M.T. with assistance from the other authors.

## Additional information

**Competing interests:** The authors declare no competing interests.

