## [Peer Review File · Nature Communications]

Reviewers' comments:

Reviewer #1 (Remarks to the Author):

The paper by Anton et al. describes an interesting new mechanism for eliminating v-src-transformed cells in both zebrafish epidermis and MDCK monolayers. They show that zebrafish EVL cells with overactive src (v-src) become eliminated during mitosis through Rho and anillin-mediated contraction. In this case, part of the cell goes apically and part goes basally. They identify that this occurs in mitosis, as inhibiting CDK1 blocks the extrusion. Moreover, they identify the key signaling that must occur for this to happen. While this is an interesting finding, several points should be addressed before publication.

Major points:

1. The debut extrusion movie (1) and in Figure 1 does not show complete extrusion, making one wonder if we are looking at a true elimination of the cell. Instead, several cells seem to pouch out and the ring contracts and then relaxes, allowing the cells to settle back into the monolayer. The other movies are also difficult to tell exactly what is going on. This needs to be demonstrated very clearly to ensure that what the authors say is happening throughout the text is really occurring.
2. Based on what the authors say is occurring—part of the cell is extruding apical and the other part basally at mitosis—the most obvious question is whether the cell is just dividing in an aberrant plane. If this is the case, it may make the findings more interesting: it could say that regulation via src is essential for keeping cells in the layer after dividing and may be an essential role for what targets it phosphorylates. This would certainly fit well with the idea its targets are critical cell polarity determinants. A paper from Jaffe and Hall had found previously that disruption of cdc42 causes cells to divide perpendicular to the plane of the epithelium in 3-D cysts, so this is consistent with the authors' findings. From all the figures and movies, I couldn't tell if the DNA was condensing or not during these 'extrusions'. What happens to the microtubule arrays during this process? It may be that src regulates the polarity of the spindle with respect to its cell-cell junctions and when overexpressed, this regulation becomes confused.
3. Is anillin found and required normally in crowding or apoptotic extrusions?
4. What happens when src is lost? Does mitosis become prolonged? It would be useful to know what occurs in the null versus the gain of function.

Minor points:

1. The images are very small in all the figures, making interpretation difficult. Levels of GFP are also blown out and difficult to interpret in many figures.
2. The discussion seems to long and rambling. If it does appear to be perpendicular cell division, the discussion should be changed accordingly. I doubt it has anything to do with Piezo1 and crowding.
3. Does the expression need to be induced or does the same thing occur if v-src is constitutively expressed?

Reviewer #2 (Remarks to the Author):

This interesting paper explores the mechanisms of oncogenic cell extrusion, a striking phenomenon which occurs when transformed cells are surrounded by normal cells. A strength of the current report is that they study the phenomenon in vivo, using the enveloping layer of the early zebrafish embryo as their model. They report fascinating relationships between oncogenic extrusion and both mitotic

regulation and cytokinesis in the transformed cell. Specifically, they show that anillin, a distinctive component of the contractile furrow, becomes recruited to the cortex of the transformed cell by p120-catenin. Further, they show that a significant proportion of cells which undergo extrusion undergo an apparently frustrated mitosis. They suggest that these first two elements, combined with aberrant Cdc42 polarity signaling may go a substantial distance to explaining the phenomenon of apically-directed oncogenic extrusion.

Overall, I think that these are potentially important findings that will be broadly informative. My principal reservation is that some key elements of the story are strongly suggestive but not developed sufficiently to be convincing. I can appreciate that the authors wish to present the potential breadth of the story, but some elements need to be strengthened to my mind.

Major questions.

1. Which of these events are cell autonomous or cell non-autonomous? Do any occur when transformed cells are grown as groups?

2. The regulation of anillin.

a) Is anillin recruited to the cortex only in cells that show mitosis-dependent extrusion or is it also recruited by the mitosis-independent pathway.

b) Functional role of anillin. The authors use a C-terminal fragment of anillin to test its functional role in cytokinesis. However, this construct binds GTP-RhoA and could potentially exert its observed effects by modulating RhoA signaling rather than by more specifically perturbing anillin. I appreciate that the mutant Rho constructs did not phenocopy the anillin mutant, but the authors' argument for anillin would be strengthened if they were to complement it with morpholinos or another dominant negative.

c) Do all extruding cells show cortical anillin or is it selective for those that show mitosis-dependent extrusion.

3. The authors suggestion that extrusion reflects a mitotic event ("splitting" of the transformed cell) is very interesting. However, I don't think that the current data (i.e. Fig 1 and Movies 1,2) establish this convincingly. The idea that the transformed cell splits is hard to appreciate from the x-y views and the pseudocoloured x-z views don't seem to capture the beginning of the process – to show that it begins with a single, whole cell. The way that the authors have presented the x-z movies, it might be as possible to interpret them as two closely-adjacent transfected cells.

I also think that the phenomenon of "splitting" needs to be quantitated: what proportion of extruding cells display splitting?

4. The p120-catenin data are intriguing, however:

a) The effects of the various mutants on cortical anillin should be quantitated.

b) I should note that the question of how p120 might be recruiting anillin is a very interesting one. Is it via protein-protein interactions (as was suggested by the Zaidel-Bar group) or by e.g. regulating RhoA? Indeed, in their discussion (line 401) the authors state that "vSrc appears to generate a narrow zone of active RhoA by modulating p120-catenin". Do they have data for this? However, I don't think that detailed mechanistic analysis is needed for this paper.

5. Fig 8. Polarity and extrusion. It is plausible to hypothesize that cell polarity may influence the direction of extrusion. Their current data to implicate Cdc42 are tantalizing, but preliminary: really only one experiment where they express a dominant-negative aPKC construct.

I'd suggest they consider localizing active Cdc42 with the WASP-RBD location biosensor. They could also reinforce the DN aPKC results with another inhibitory strategy, such as co-injecting Cdc42 shRNA/morpholinos or even the WASP-RBD at higher levels.

6. Fig 8b,c. The authors reconstitute v-Src-induced extrusion by manipulating multiple mechanisms. However, while I appreciate that this is an heroic experiment, the data shown – essentially one movie – are anecdotal. At the least, they should:

- a) quantitate the proportion of expressing cells that undergo extrusion; and
- b) exclude the possibility that cells are extruding because they are undergoing apoptosis (as a consequence of expressing the 4 transgenes).

7. Fig 6B. I'm sorry, but I don't understand how the authors have performed the assay or what the data are meant to show. Presumably they have expressed the mammalian Cdk1 peptide and stained with the mammalian antibody? If so, why is the signal lower in the Src cells?

Tiny points

Line 53. "Cell extrusion". At least phenomenologically, extrusion is a diverse process. Just for absolutely clarity, I think it would be good for the authors to always be explicit that they are dealing with oncogenic extrusion.

Responses to reviewers:

We thank the reviewers for their constructive comments, which we found very helpful. We have considered the issues raised by the reviewers, and made substantial changes in the revised manuscript. At the end of this letter, we added “Summary-of-changes”, including additional Figures, which were necessary to optimise the format for Nature Communications.

Reviewer #1 (Remarks to the Author):

The paper by Anton et al. describes an interesting new mechanism for eliminating v-src-transformed cells in both zebrafish epidermis and MDCK monolayers. They show that zebrafish EVL cells with overactive src (v-src) become eliminated during mitosis through Rho and anillin-mediated contraction. In this case, part of the cell goes apically and part goes basally. They identify that this occurs in mitosis, as inhibiting CDK1 blocks the extrusion. Moreover, they identify the key signaling that must occur for this to happen. While this is an interesting finding, several points should be addressed before publication.

First, we thank this reviewer for insightful comments and for recognizing our paper with an interesting new mechanism. We provided quantitative data to support our findings and improved imaging quality.

Major points:

1. The debut extrusion movie (1) and in Figure 1 does not show complete extrusion, making one wonder if we are looking at a true elimination of the cell. Instead, several cells seem to pouch out and the ring contracts and then relaxes, allowing the cells to settle back into the monolayer. The other movies are also difficult to tell exactly what is going on. This needs to be demonstrated very clearly to ensure that what the authors say is happening throughout the text is really occurring.

We have changed the initial movie to show multiple events of complete extrusion. We have also changed Movie2 with the xz view of apico-basal extrusion and correlated it with the first movie (it now shows the cell indicated as “1” in Movie1). Accordingly, Fig. 1B and Fig. 1C and relevant Movies 1 and 2 with xy and xz movies of vSrc-driven extrusion were changed to better illustrate this process.

2. Based on what the authors say is occurring—part of the cell is extruding apical and the other part basally at mitosis—the most obvious question is whether the cell is just dividing in an aberrant plane. If this is the case, it may make the findings more interesting: it could say that regulation via src is essential for keeping cells in the layer after dividing and may be an essential role for what targets it phosphorylates. This would certainly fit well with the idea its targets are critical cell polarity determinants. A paper from Jaffe and Hall had found previously that disruption of cdc42 causes cells to divide perpendicular to the plane of the epithelium in 3-D cysts, so this is consistent with the authors’ findings. From all the figures and movies, I couldn’t tell if the DNA was condensing or not during these ‘extrusions’. What happens to the microtubule arrays during this process? It may be that src regulates the polarity of the spindle with respect to its cell-cell junctions and when overexpressed, this regulation becomes confused.

After extensively investigating what happens to chromatin (Fig. 3E,F), the nucleus (S4E) and microtubules during extrusion (Fig. S4D) as well as DNA and some components of the nuclear envelope (data not shown), we have concluded that this is not the case and Src cells are not just dividing in an aberrant plane. The nucleus appears largely intact (apart from apparent leakiness of the envelope visible in Fig. 3E and S4E) and microtubules do not form a mitotic spindle. However, the cytokinetic machinery appears to be used to perform the split, as demonstrated in Fig. 2. Importantly, our key finding is that vSrc is capable of coupling the cytokinetic machinery

with cell junctions and of fine-tuning apical polarity in early mitosis, allowing apicobasal split during extrusion. This was uncovered only by live-imaging in this study.

We have done several additional experiments to clarify the role of apical polarity in vSrc cell extrusion and presented in current Figure 8. The details are described in the point 5 to Reviewer 2.

3. Is Anillin found and required normally in crowding or apoptotic extrusions?

We cannot perform crowding experiments in the EVL until 16h post-fertilisation (hpf) when we did all the analyses for cell extrusion in this paper. Until 24hpf, the EVL is a simple squamous epithelial sheet surrounding the yolk under considerable tension due to the curvature of the embryo. We have never observed crowding in this context. However, by 48 hpf, the EVL becomes two-layered and the EVL cells are eventually replaced by embryonic epidermis by unknown mechanisms. Indeed, crowding-induced extrusion is observed in the two-layered epithelium of 48hpf embryos (Eisenhoffer et al., 2012). Although this is a very interesting question, it is beyond the scope of this paper, as epithelial contexts are different.

In terms of apoptotic extrusion, we were able to address the question about Anillin ring during cell death by expressing DAPK1 (as in Fig. S2D). We observed that most extrusions occur without prior cortical recruitment of Anillin (Referee Fig. 1A). Occasionally, DAPK1-induced extrusion occurs immediately following cell division, when a portion of Anillin is located cortically (Referee Fig. 1B). On closer inspection, cortical Anillin does not appear to form a stable junctional ring and it seems to fall off the membrane, so it is unlikely to be required for this process.

Referee Fig. 1. Stable Anillin ring is not assembled during extrusion due to cell death. (A, B) Time-lapse imaging of Anillin-GFP during DAPK1-driven basal cell extrusion. Embryos were injected with the following constructs: dUAS:myr-Cherry;DAPK1 and Krt18:Anillin-GFP. Movies were taken over 4 hours. Frames were extracted from representative movies at indicated times from the tailbud stage (t=0). Scale bars, 20 μ m. **(A)** Cell death-induced extrusion without cortical Anillin. **(B)** Cell death-induced extrusion following cell division with residual Anillin at the cortex.

4. What happens when src is lost? Does mitosis become prolonged? It would be useful to know what occurs in the null versus the gain of function.

Addressing this intriguing question is unfortunately beyond the scope of our paper. Our aim was not to elucidate the endogenous function of the Src kinase, but rather focus on what happens when Src is mutated in the context of carcinogenesis. Crucially, we do not currently have the tools to perform transient knockout of a gene in a mosaic manner. Morpholinos provide a knockout in the whole embryo and the CRISPR technology is not yet reliable enough to use it transiently, although we are currently trying to optimise it for future use.

Minor points:

1. *The images are very small in all the figures, making interpretation difficult. Levels of GFP are also blown out and difficult to interpret in many figures.*

We have increased the size of most images. In our experiments, levels of GFP increase over time and during extrusion, as vSrc and other markers are expressed in an inducible fashion from UAS elements. This makes it hard to control brightness throughout movies. We tried to optimise the settings to illustrate our points as much as possible, but would be happy to correct any specific figures, which were unacceptable to the reviewer.

2. *The discussion seems to long and rambling. If it does appear to be perpendicular cell division, the discussion should be changed accordingly. I doubt it has anything to do with Piezo1 and crowding.*

The discussion was shortened and amended. In particular, we totally removed the paragraph describing Piezo.

3. *Does the expression need to be induced or does the same thing occur if v-src is constitutively expressed?*

As we work in embryos, it is difficult to create a fully constitutive system (promoters switch on at certain stages of development). For instance, Krt18 promoter that we use is fully turned on at about 30% epiboly. Our KalTA4-ERT2 system gives us additional control and allows us to induce vSrc expression with tamoxifen between 50-70% epiboly. To address the reviewer's question, we cloned vSrc directly under Krt18 promoter without the need of external induction (Referee Fig. 2). In this experiment, although the number of cells per embryo generally increased ($n_{\text{Krt18:vSrc}} = 32$, $n_{\text{UAS:vSrc}} = 24$) reflecting the longer expression of vSrc, extrusion rates remained very similar: "apical extrusion" was 8.9% as compared to 7.6% for the UAS:vSrc, while "apical extrusion and rounding" was 20% as compared to 17.5% for the UAS:vSrc. Along with the Fig. S2B, that also suggests that, at least in the fish embryos, vSrc-mediated extrusion is predominantly an autonomous event independent of cell density or group size.

Referee Fig. 2. Extrusion rates are comparable when vSrc is expressed directly under Krt18 promoter without induction. Quantification of vSrc cell extrusion and rounding rates based on fixed embryos. Embryos were injected with the Krt18:EGFP-vSrc. Data represent one experiment (number of embryos: $n_{\text{Src}}=12$; total number of cells: 380)

Reviewer #2 (Remarks to the Author):

This interesting paper explores the mechanisms of oncogenic cell extrusion, a striking phenomenon which occurs when transformed cells are surrounded by normal cells. A strength of the current report is that they study the phenomenon in vivo, using the enveloping layer of the early zebrafish embryo as their model. They report fascinating relationships between oncogenic extrusion and both mitotic regulation and cytokinesis in the transformed cell. Specifically, they show that anillin, a distinctive component of the contractile furrow, becomes recruited to the cortex of the transformed cell by p120-catenin. Further, they show that a significant proportion of cells which undergo extrusion undergo an apparently frustrated mitosis. They suggest that these first two elements, combined with aberrant Cdc42 polarity signaling may go a substantial distance to explaining the phenomenon of apically-directed oncogenic extrusion.

Overall, I think that these are potentially important findings that will be broadly informative. My principal reservation is that some key elements of the story are strongly suggestive but not developed sufficiently to be convincing. I can appreciate that the authors wish to present the potential breadth of the story, but some elements need to be strengthened to my mind.

First, we thank this reviewer for appreciating our work and for providing critical and constructive comments. Now, we strengthened the issue regarding how apical polarity signalling is involved in vSrc-driven extrusion, and also added quantitative data to support our findings.

Major points:

1. Which of these events are cell autonomous or cell non-autonomous? Do any occur when transformed cells are grown as groups?

Quantifying extrusion according to the group size of vSrc cells in zebrafish embryos is quite challenging, as we usually only image a part of the embryo due to its curvature. To address this question, we assumed that larger group size would correlate with higher density of vSrc cells visible in our images and estimated extrusion rates depending on cell density (Fig. S2B). In this quantification extrusion and rounding rates of vSrc cells remain constant, therefore we propose that for vSrc the described events are predominantly autonomous. Moreover, while inhibiting the cell cycle in the whole embryo as compared to in vSrc cells only (Fig. 3B, S4A and B), we observe similar levels of blocking extrusion. Consistent with this notion, in case of Anillin or cell polarity, autonomous inhibition by dominant negative versions is sufficient to impair extrusion.

2. The regulation of anillin.

a) Is anillin recruited to the cortex only in cells that show mitosis-dependent extrusion or is it also recruited by the mitosis-independent pathway?

This is a very good question, which actually led us to some clarifications about previous experiments we presented. We have claimed in the previous version that G1 arrest of vSrc cells, by using p20, p21 or inhibitors HU/Aph, does not affect extrusion. Our previous data have shown that in wild type embryos these treatments are sufficient to arrest cells in G1. In the context of vSrc, however, it turns out that this is not the case. Imaging embryos with vSrc and p21 on the Ccnb1-GFP background revealed that vSrc cells are still able to progress to the G2 (accumulate CcnB1). Presumably vSrc dominates over the G1 block, likely though activating Pp2a. These findings prompted us to remove the p21, p20 and HU/Aph data from the paper.

Despite this fact, the conclusion that G1 extrusion can occur remains valid, as we saw it in live imaging with Ccnb1-GFP and vSrc in the embryo (Fig. 4D) as well as in the tissue culture system (Fig. 5C).

Answering to the reviewer's question about the involvement of Anillin in cell cycle-independent extrusion, we could not simultaneously image Anillin-GFP and the Ccnb1-GFP marker. Cell cycle-

independent extrusion is a rare event, difficult to capture, so without Ccnb1-GFP or enrichment for vSrc cells arrested in G1, we could not be certain it occurred.

Instead, we have decided to use a constitutively active aPKC (myr-aPKC, explained in more details below) that induces purely apical extrusion, but not apicobasal extrusion, on its own without vSrc. We have shown before that this type of extrusion is independent of the cell cycle, as in contrast to vSrc, blocking G2/M with the Emi1 morpholino does not inhibit myr-aPKC-mediated extrusion (data not shown). We believe that this type of extrusion is driven purely by apical domain expansion (see Fig. 8A), as myr-aPKC localises to the entire cell membrane, changing its identity to apical. We have therefore imaged Anillin-GFP in this type of extrusion and found no recruitment of Anillin to the junctions or cortex (see Referee Fig 3, for example).

Referee Fig. 3. Anillin ring is not assembled during cell cycle-independent myr-aPKC-driven extrusion. (A, B) Time-lapse imaging of Anillin-GFP during myr-aPKC-driven apical cell extrusion. Embryos were injected with the following constructs: dUAS:myr-Cherry;myr-aPKC and Krt18:Anillin-GFP. Movies were taken over 4 hours. Frames were extracted from representative movies at indicated times from the tailbud stage (t=0). Scale bars, 20 μ m. (A, B) are examples of different extrusion event from independent experiments.

b) Functional role of anillin. The authors use a C-terminal fragment of anilin to test its functional role in cytokinesis. However, this construct binds GTP-RhoA and could potentially exert its observed effects by modulating RhoA signaling rather than by more specifically perturbing anillin. I appreciate that the mutant Rho constructs did not phenocopy the anillin mutant, but the authors' argument for anillin would be strengthened if they were to complement it with morpholinos or another dominant negative.

To address this question we have performed a rescue experiment with the Anillin morpholino. Since this morpholino delays the process of epiboly, we could not compare extrusion rates between Anillin morpholino- and control morpholino-treated embryos. However, we compared embryos treated with Anillin morpholino and injected with either vSrc only or vSrc and morpholino-resistant Anillin. This “rescue” experiment revealed that without Anillin extrusion rates are lower than in the presence of Anillin expressed autonomously (Fig. 2D). Fig. 2D with the effect of the presence of Anillin on vSrc-mediated extrusion was added to prove that Anillin is necessary for extrusion.

c) Do all extruding cells show cortical anillin or is it selective for those that show mitosis-dependent extrusion.

The answer to this question was included in Point 2 a). Please see above.

3. The authors suggestion that extrusion reflects a mitotic event (“splitting” of the transformed cell) is very interesting. However, I don’t think that the current data (i.e. Fig 1 and Movies 1,2) establish this convincingly. The idea that the transformed cell splits is hard to appreciate from the x-y views and the pseudocoloured x-z views don’t seem to capture the beginning of the process – to show that it begins with a single, whole cell. The way that the authors have presented the x-z movies, it might be as possible to interpret them as two closely-adjacent transfected cells. I also think that the phenomenon of “splitting” needs to be quantitated: what proportion of extruding cells display splitting?

We have changed figures 1B and 1C as well as corresponding movies 1 and 2 and correlated them: cell marked in 1B as “1” is the cell shown in 1C in xz view.

Based on live imaging, 62% of Src cells display cell splitting with large (at least 1/3 of the original cell) basal parts produced (Fig. S2A). The remaining 38% produces small basal vesicles during extrusion.

4. The p120-catenin data are intriguing, however:

a) The effects of the various mutants on cortical anillin should be quantitated.

This quantification proved quite challenging as Anillin behaves very differently in the presence of either form of p120-catenin. With p120-mutFF, Anillin forms large clusters, which often fall out of the membrane (Fig. 7F). In the presence of p120-mutEE, Anillin is more stable at the cortex, but the fluorescence intensity of Anillin is considerably weaker and there is no indication of any clusters forming (Fig. 7D, Movie 9). We attempted to quantify this by automatically detecting the brightest Anillin spots using the “Spots” function of the Imaris software. We then averaged the intensity of detected spots per cell for each of three consecutive time points as well as their position in regards to the cortex (away or within the cortex) and presented these results in figures S7C and D. We appreciate that this is a rather crude quantification, but we hope that it illustrates our point: p120-mutFF does not stably recruit Anillin to the cortex, while p120-mutEE, does. However, we realise that cortical localisation of Anillin is not as efficient with p120-mutEE compared to vSrc (as suggested by the low intensity of Anillin at the cortex). We believe that vSrc may promote cortical Anillin recruitment via modifying more residues within p120-catenin or perhaps through other proteins. However, we managed to show that the two residues within p120 are indeed crucial for the recruitment. We also show that p120-mutEE acts as a factor promoting apicobasal split in our reconstitution experiments, so at this level of Anillin recruitment appears to be sufficient (Fig 9A,B).

b) I should note that the question of how p120 might be recruiting anillin is a very interesting one. Is it via protein-protein interactions (as was suggested by the Zaidel-Bar group) or by e.g. regulating RhoA? Indeed, in their discussion (line 401) the authors state that “vSrc appears to generate a narrow zone of active RhoA by modulating p120-catenin”. Do they have data for this? However, I don’t think that detailed mechanistic analysis is needed for this paper.

It has been previously shown that the vSrc phosphorylation sites on p120-catenin, promoting Anillin recruitment, regulate the interaction between p120 and RhoA. Therefore, we speculated that Anillin recruitment occurs via RhoA. However, at this stage we cannot conclude whether this happens simply due to direct protein-protein interactions or if vSrc also regulates RhoA activity that in turn plays a role in this process. At the same time, we have shown that precise control of the RhoA activity plays a role in rounding prior to extrusion (Fig. 7A), but RhoA activation on its own is not sufficient to drive extrusion or rounding (Fig. S7A). We hope that we will be able to answer this

question in the future, as we are planning to create some optogenetic tools for RhoA activation in the embryo. However, this is beyond the scope of this manuscript.

5. Fig 8. Polarity and extrusion. It is plausible to hypothesize that cell polarity may influence the direction of extrusion. Their current data to implicate Cdc42 are tantalizing, but preliminary: really only one experiment where they express a dominant-negative aPKC construct. I'd suggest they consider localizing active Cdc42 with the WASP-RBD location biosensor. They could also reinforce the DN aPKC results with another inhibitory strategy, such as co-injecting Cdc42 shRNA/morpholinos or even the WASP-RBD at higher levels.

Following the reviewer's and Editor's suggestions, we have done several additional experiments to clarify the role of apical polarity in vSrc cell extrusion. The main experiments are as follows; (1) we inhibited Cdc42 with overexpressed WASP-RBD domain (Fig. 8D), (2) we expressed a dominant negative version of Par3 (Fig. 8C) and (3) we mutated a site in aPKC that can potentially be directly phosphorylated by vSrc (Fig. 8E). All of these experiments resulted in inhibition of vSrc cell extrusion, thus implying that the polarity-dependent, rather than polarity-independent, role of aPKC is important for this process.

In addition, as requested by the reviewer, we have imaged WASP-RBD-GFP as a Cdc42 sensor in vSrc cells, but unfortunately we could not see it clearly localising to the junctions (Referee Fig. 4). Since previously we also attempted to image RhoA (by using Rhothekin-RBD-GFP domain) and were unable to see enrichment even during cytokinesis in the cleavage furrow, we believe these sensors may not be sensitive enough to visualise active GTP-ases in the EVL using live-imaging. Given that the amount of active Cdc42 or RhoA does not have to be large in order for them to have an effect, it is likely that the sensors lack sensitivity to visualise these two GTPases participating in extrusion.

We think that Cdc42 plays roles in vSrc extrusion based on the data presented in this paper, other literature (cited in the paper) and our own unpublished results. As mentioned earlier, membrane-bound myr-aPKC alone induces apical extrusion (42% of cells in live imaging become extruded over 6 hours, data not shown). Likewise, constitutively active Cdc42 (Cdc42-Q61L) is sufficient to induce extrusion (Referee Fig. 4C), although it is not as potent as membrane bound aPKC and we have never managed to capture that extrusion using live-imaging. That is why we speculated that aPKC might also be directly regulated by vSrc in this context. This point was extended in Discussion in Page10.

Referee Fig. 4. Localization of the Cdc42 sensor in vSrc-mediated extrusion. (A, B) Time-lapse imaging of WASP-RBD-GFP during vSrc-driven cell extrusion. Embryos were injected with the following constructs: dUAS:myr-Cherry-vSrc and Krt18:WASP-RBD-GFP. Movies were taken over 4 hours. Frames were extracted from representative movies at indicated times from the tailbud stage ($t=0$). Scale bars, 25 μm . (A, B) are examples of different extrusion event from independent experiments. **(C)** Quantification of Cdc42-Q61L-mediated cell extrusion and rounding rates based on fixed embryos. Embryos were injected with the UAS:GFP-Cdc42-Q61L construct. Data represent 4 independent experiments (number of embryos: $n_{\text{Cdc42}}=31$).

6. Fig 8b,c. The authors reconstitute v-Src-induced extrusion by manipulating multiple mechanisms. However, while I appreciate that this is an heroic experiment, the data shown – essentially one movie – are anecdotal. At the least, they should:

a) quantitate the proportion of expressing cells that undergo extrusion; and

We attempted to quantify the initial reconstitution experiment and realised that most extrusions induced with myr-aPKC as a polarity factor were apical, not apicobasal (Referee Fig. 5). We speculated that the activity of myr-aPKC was too strong in relation to other used modulators and dominated in this experiment. We then attempted to modulate aPKC activity by either changing its expression levels or using different active forms. The most efficient was an active form of aPKC (delN-aPKC), which lacks the N-terminal regulatory domain. Despite the fact that on its own delN-aPKC did not induce apical extrusion, it reliably stimulated apicobasal split in the company of other vSrc-mimicking factors (wee1, cdc25, XIAP and p120EE)(Fig 9A, B and quantification in Fig. S8B).

b) exclude the possibility that cells are extruding because they are undergoing apoptosis (as a consequence of expressing the 4 transgenes).

The G2/M arrest-dependent basal extrusions were highly associated with cell death (numerous basal vesicles produced), but the apicobasal splits observed in the reconstitution experiments were not related to cell death. This was supported by the data that no Cleaved-Cas3 staining could be

observed in large basal parts of cells or cells undergoing apicobasal extrusion in these embryos (Fig. S8C). Therefore, we replaced the previous Fig 8 with the current Fig 9.

Referee Fig. 5. Expression of myr-aPKC in the EVL results in apical extrusion that can be partly converted into apico-basal extrusion in the presence of vSrc-activity-mimicking factors. (A, B) Quantification of myr-aPKC-driven (A) and (p120EE, myr-aPKC, wee1, cdc25, XIAP)-driven (B) cell extrusion type based on time-lapse imaging. 7-8 embryos were imaged per condition in 7 independent experiments (total number of extrusions: $n_{\text{myr-aPKC}} = 27$, $n_{\text{p120EE,myr-aPKC,wee1,cdc25,XIAP}} = 23$).

7. Fig 6B. I'm sorry, but I don't understand how the authors have performed the assay or what the data are meant to show. Presumably they have expressed the mammalian Cdk1 peptide and stained with the mammalian antibody? If so, why is the signal lower in the Src cells?

Since it was not possible to detect endogenous phospho-CDK1 in the embryo, we expressed a CDK-consensus peptide with a red fluorescent tag together with vSrc in the EVL. We then stained for phospho-CDK1 with an antibody. Indeed it might be confusing that the picture without vSrc seems brighter, but in fact this is non-specific signal, while in the vSrc cell nucleus is brightly stained. We have outlined the shape of the GFP and vSrc cell in Fig 6B to make it clearer. This was added to M&M clearly under "Immunostaining of fish embryos".

Minor points

Line 53. "Cell extrusion". At least phenomenologically, extrusion is a diverse process. Just for absolutely clarity, I think it would be good for the authors to always be explicit that they are dealing with oncogenic extrusion.

This was corrected as suggested.

Changes-of-summary

Changes to the main text

1. Fig. 1B and Fig. 1C and relevant Movies 1 and 2 with xy and xz movies of vSrc-driven extrusion were changed to better illustrate this process.
2. Fig. 2D with the effect of the presence of Anillin on vSrc-mediated extrusion was added to prove that Anillin is necessary for extrusion.
3. The original Fig. 3E with the effect of p20,p21 on vSrc-induced extrusion was removed, as we found during Ccnb1 imaging that expression of these factors did not result in a G1 block in the context of vSrc.
4. Fig. 7D and Movie 9 representing recruitment of Anillin by p120-mutEE was changed, as we realised that the initial imaging was performed in the presence of LifeAct-RFP, which affects recruitment of Anillin to the cortex. The new imaging and quantification was done in wild type embryos for both p120-EE and p120-FF forms.
5. Fig. 8A with a scheme of how Cdc42 regulates apical polarity was added. Fig. 8C, D and E with the effects of DN-Par3, WASP-RBD-GFP (titrating Cdc42), and phosphomimetic aPKC-Y417F on vSrc-mediated extrusion were added. These figures serve to support and clarify the role of polarity in vSrc-driven extrusion.
6. Fig. 9A, 9B and Movies 10 and 11 with xy and xz views of reconstitution of vSrc-like extrusion event without the oncogene were changed. We found that using a different form of active aPKC, aPKC-delN instead of myr-aPKC, gave us much more reliable reconstitution of apicobasal extrusion.

Changes to supplementary material

1. Movies: 1, 2, 9, 10 and 11 were replaced (see reasons above).
2. Fig. S2A with quantification of the type of extrusion (either apical or apicobasal) following vSrc induction was added.
3. Fig. S2B with quantification of vSrc-driven cell extrusion and rounding rates depending on the density of transformed cells was added. This supports the claim that this type of extrusion is predominantly autonomous.
4. The original Fig. S2D and S2E with the effect of G1 inhibitors HU/Aph on vSrc-induced extrusion were removed, as we found during Ccnb1 imaging that using these inhibitors did not result in a G1 block in the context of vSrc.
5. Fig. S7C and S7D with quantification of Anillin recruitment to the cortex with either p120-mutEE or p120-mutFF were added. These figures show that automatically detected spots of Anillin in cells with a cell cycle block and p120-mutFF are brighter and often fall off the cortex (are not stably recruited), while with p120-mutEE are more spread out and predominantly localise to the cortex.
6. Fig. S8B with quantification of reconstitution of vSrc-like extrusion was added.
7. Fig. S8C with Cleaved-Caspase-3 staining of reconstitution of vSrc-like extrusion was added to prove that vSrc-like cells following apicobasal split remain alive and that this type of extrusion does not occur following cell death.

REVIEWERS' COMMENTS:

Reviewer #2 (Remarks to the Author):

The authors have made extensive efforts to address the questions that I raised in my original review. I appreciate their good faith and that they were often pushing the limits of what is technically possible. Overall, I think that they have reasonably addressed my questions. This is an interesting report and it will be good for the field to see it.

Reviewer #3 (Remarks to the Author):

Anton et al. have taken steps to adequately address the suggestions of the reviewers and the revised version of the manuscript "Src-transformed cells hijack mitosis to extrude from the epithelium" is now suitable for publication. A few minor points should be addressed prior to publication.

1. The authors nicely show that phosphorylation of p120-catenin by vSrc was necessary for apicobasal vSrc-cell extrusion. Yet in mutants that mimic a permanent state of phosphorylation, it is difficult to determine the basal extrusion event (Figure 7D and Movie 9 (pg7 line 304 and pg8 line 322) based on the data as presented. The addition of a X/Z projection for Figure 7 D-F would help the reader follow the ejected cell in the apical or basal direction. It would also be useful to provide quantification of the number of apical vs basal extrusion events under these conditions.

2. In Figure 4C, arrowheads should be added to denote the localization of ccB1-GFP and the changes induced by vSrc.

3. There is an aberrant white bar over the text label in Figure 5B.

Response to reviewers

Reviewer #2 (Remarks to the Author):

The authors have made extensive efforts to address the questions that I raised in my original review. I appreciate their good faith and that they were often pushing the limits of what is technically possible. Overall, I think that they have reasonably addressed my questions. This is an interesting report and it will be good for the field to see it.

Thank you very much for critically reading and assessing our report. We appreciate your positive opinion and are glad that we were able to adequately address your concerns.

Reviewer #3 (Remarks to the Author):

Anton et al. have taken steps to adequately address the suggestions of the reviewers and the revised version of the manuscript "Src-transformed cells hijack mitosis to extrude from the epithelium" is now suitable for publication. A few minor points should be addressed prior to publication.

Thank you for reviewing our manuscript and great comments.

1. The authors nicely show that phosphorylation of p120-catenin by vSrc was necessary for apicobasal vSrc-cell extrusion. Yet in mutants that mimic a permanent state of phosphorylation, it is difficult to determine the basal extrusion event (Figure 7D and Movie 9 (pg7 line 304 and pg8 line 322) based on the data as presented. The addition of a X/Z projection for Figure 7 D-F would help the reader follow the ejected cell in the apical or basal direction. It would also be useful to provide quantification of the number of apical vs basal extrusion events under these conditions.

Figure 7E was added depicting an xz view of extrusion from Figure 7D, as requested. As mentioned in the manuscript these types of extrusions were only occasional, in 5 movies we observed them twice and both were basal. We have modified the test to highlight this information.

2. In Figure 4C, arrowheads should be added to denote the localization of ccB1-GFP and the changes induced by vSrc.

Arrowheads were added in figures 4A and 4B as suggested.

3. There is an aberrant white bar over the text label in Figure 5B.

We corrected the position of scale bars in this figure.